# ALPS: ADAPTIVE LLM PRUNING VIA GRADIENT SEARCH IN LEARNED REPRESENTATION SPACE

## ABSTRACT

Deploying Large Language Models (LLMs) at the edge is crucial for data privacy and offline operation, yet their massive parameter count poses significant resource challenges. While existing methods rely on discrete-space heuristics to search for pruning configurations, we introduce a fundamentally different approach: reformulating the search for optimal LLM pruning configurations as gradient optimization in a learned continuous representation space. Our method, `Alps` (Adaptive Layer Pruning via Search), embeds discrete pruning configurations into a continuous space where efficient gradient-based optimization becomes possible, then decodes optimal representations back to implementable discrete pruning schemes. This encoder-evaluator-decoder architecture automatically learns from collected "pruning-score" data pairs, eliminating manual tuning while jointly optimizing for model performance, latency, and energy consumption in a deployment-specific manner. Extensive experiments across Llama-7B, Llama2-7B, Llama2-13B, and Vicuna-7B demonstrate `Alps`'s superiority, achieving up to 34.1% energy reduction and 33.5% lower latency while maintaining over 91% of original performance. At high pruning ratios (50%), `Alps` consistently outperforms state-of-the-art methods in both perplexity and downstream task accuracy.

## 1 INTRODUCTION

Large Language Models (LLMs) (Chowdhery et al., 2023; OpenAI, 2023) have transformed AI with their capacity to comprehend and generate human language (Thirunavukarasu et al., 2023; Singhal et al., 2022). While primarily cloud-based, deploying LLMs to edge devices—PCs, smartphones, and IoT systems—is gaining traction (Apple, 2024; Singh et al., 2023). Edge deployment enhances privacy and enables offline operation, critical for applications like conversational agents (OpenAI, 2023), search engines (microsoft), and coding assistants (github). However, the massive parameter count creates significant deployment challenges. For instance, Llama-7B (Touvron et al., 2023) requires approximately 14GB of memory in 16-bit precision, far exceeding the typical 4-12GB available on edge devices (Statista Inc., 2021). These memory and computational constraints make effective compression essential for practical edge deployment of LLMs.

Traditional CNN/DNN pruning techniques (Yang et al., 2017; Molchanov et al., 2016) fail for LLMs, degrading generation capabilities due to fundamental differences in network architecture and auto-regressive inference patterns. Recent LLM-specific approaches (Ma et al., 2023; Men et al., 2024; Ashkboos et al., 2024) maintain generative functionality while reducing model size through layer pruning (Men et al., 2024) or block pruning (Ma et al., 2023; Ashkboos et al., 2024). Yet these methods focus primarily on model size reduction, with latency and energy improvements being passive byproducts of reduced computation rather than explicit optimization objectives—factors critical for practical edge deployment. Furthermore, they rely on discrete-space heuristics and manual intervention, making exhaustive exploration of configuration spaces computationally prohibitive. These methods produce "one-size-fits-all" solutions that cannot adapt to specific hardware constraints or deployment requirements—a critical limitation when models must run on diverse platforms, from resource-constrained smartphones to edge servers with different performance priorities.

This need for hardware-specific optimization fundamentally clashes with the discrete search paradigm of existing methods. Finding optimal configurations for different deployment contexts requires exploring an intractably large space where the combinatorial explosion of layer-wise pruning options

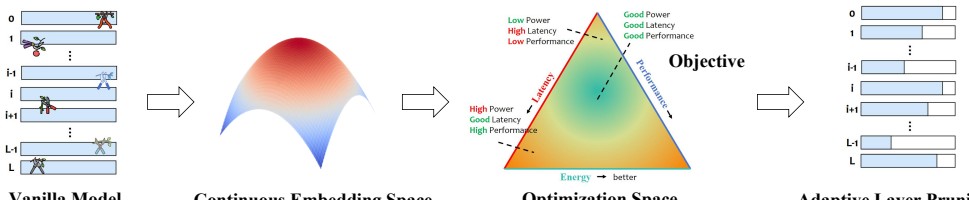

Figure 1: Overview of `Alps`: transforming discrete pruning configuration search into continuous-space gradient optimization, enabling joint optimization of model performance, latency, and energy.

makes exhaustive evaluation prohibitive. We ask: ***Can we reformulate this discrete search as gradient optimization in a learned continuous representation space?*** Mapping discrete configurations into a continuous space would enable efficient gradient-based discovery of device-specific solutions.

In this paper, we introduce `Alps` (Adaptive Layer Pruning via Search), a framework that reformulates the search for optimal LLM pruning configurations as a gradient optimization problem in a learned continuous space (as depicted in Figure 1). Unlike discrete search methods that directly manipulate pruning configurations, `Alps` embeds discrete pruning configurations into a continuous representation space. This enables efficient gradient-steered optimization for optimal pruning representation before decoding back to implementable discrete formats. This approach overcomes the combinatorial explosion of configuration options that makes exhaustive search impractical in discrete spaces. Our approach comprises four key steps: (1) leveraging conventional random and heuristic pruning methods to automatically curate a robust training dataset of "pruning-score" pairs; (2) employing an encoder-evaluator-decoder architecture that jointly optimizes sequence-to-sequence (Sutskever et al., 2014) and score estimation losses on the collected data to learn a continuous representation space; (3) applying gradient-steered optimization within this space to pinpoint the optimal pruning representation; and (4) utilizing beam search (Freitag & Al-Onaizan, 2017) to generate the final optimal pruning configuration from the optimal representation via the trained decoder. The key insight is that while discrete pruning spaces suffer from combinatorial explosion, we can learn a continuous representation that encodes the essential structure of effective pruning strategies, enabling powerful gradient-based methods to discover configurations beyond the reach of discrete search.

In summary, `Alps` framework offers significant advantages over conventional LLM pruning approaches: (1) A novel continuous-space formulation that transforms intractable discrete search into efficient gradient optimization, achieving up to 8.15% performance improvement over state-of-the-art heuristic methods while eliminating manual tuning. (2) Joint multi-objective optimization of model performance, latency, and energy consumption, delivering up to 34.1% energy reduction and 33.5% latency improvement while maintaining over 91% of original model performance. (3) Adaptability across model scales and architectures, with consistent performance gains over state-of-the-art methods. Effectiveness is maintained even at higher pruning ratios (50%), showing robust improvements in perplexity and downstream task accuracy across diverse deployment scenarios.

## 2 DEFINITIONS AND PROBLEM STATEMENT

We present a generalized and efficient generative LLM pruning framework tailored for resource-constrained edge devices. Our approach targets a set of candidate pruning layers, $L = [l_1, l_2, \ldots l_k]$, for $k$ decoder layers of LLMs like 32-layer Llama-7B (Touvron et al., 2023). Using classic pruning algorithms, we collect training data comprising $n$ "pruning-score" pairs, $\mathcal{R} = \{(\mathcal{P}_i, s_i)\}_1^n$, where each configuration $\mathcal{P}_i = [r_1, \ldots, r_k]$ represents layer-wise pruning ratios. To make the decoding problem tractable, we define each ratio $r_i$ from a predefined discrete vocabulary $\mathcal{C} = \{0.00, 0.01, \ldots, 1.00\}$ containing 101 candidate values. This design transforms our decoder's task into a standard sequence generation problem, where each step selects a ratio from $\mathcal{C}$ rather than regressing to an unbounded continuous value. The term $s_i$ denotes the comprehensive performance score. Unlike prior works (Ma et al., 2023; Men et al., 2024) focusing solely on performance and efficiency, we concurrently evaluate: (1) *good generation ability*; (2) *low inference latency*; (3) *low energy consumption* to characterize $s_i$ for a given $\mathcal{P}_i$, ensuring a balanced optimization for resource-constrained edge devices. The holistic metric function, $\mathcal{F}$, evaluates the effectiveness of various pruning configurations, $\mathcal{P}_i$:

$$s_i = \mathcal{F}(\mathcal{P}_i) = \frac{P_e \, P_t}{P_g} \tag{1}$$

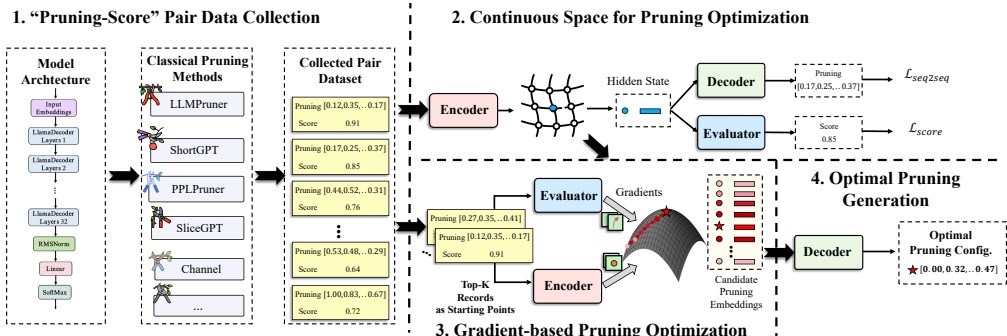

Figure 2: An overview of `Alps` framework.

where $P_g$, *i.e.* the zero-shot perplexity, quantifies the generative capabilities, with lower values indicating more precise model predictions. $P_t = (\frac{T}{t_i})^{\mathbf{1}(t_i > T)\beta}$ and $P_e = (\frac{E}{e_i})^{\mathbf{1}(e_i > E) \times \alpha}$, where $T$ and $E$ denote the latency and energy budgets for edge devices, respectively, specified by the developer. $e_i$ and $t_i$ represent the energy consumption and inference latency for a given pruning ratio $\mathcal{P}_i$. The indicator function $\mathbf{1}(x)$ returns 1 if condition $x$ holds, and 0 otherwise. LLMs under certain pruning ratios that exceed the desired latency and energy constraints will be penalized by developer-specified factors $\alpha$ and $\beta$, both set to 2 in our implementation. We aim to learn a continuous representation space $\Theta$ for "pruning-score" pairs. Three key modules are proposed: (1) An encoder $q$ that maps a pruning configuration $\mathcal{P}_i$ to its continuous representation $\mathcal{E}_{\mathcal{P}_i}$; (2) An evaluator $e$ that scores a pruning representation $\mathcal{E}_{\mathcal{P}_i}$ via $s_i = e(\mathcal{E}_{\mathcal{P}_i})$; (3) A decoder $g$ that reconstructs the discrete pruning configuration $\mathcal{P}_i$ via $\mathcal{P}_i = g(\mathcal{E}_{\mathcal{P}_i})$. These modules are jointly optimized on the collected dataset $\mathcal{R}$ of "pruning-score" pairs. With the learned $\Theta$, $q$, $e$ and $g$, we can perform gradient-steered search in $\Theta$ to identify the optimal pruning configuration $\mathcal{P}^*$, defined as:

$$\mathcal{P}^* = g\left(\mathcal{E}_{\mathcal{P}^*}\right) = g(\text{argmax}_{\mathcal{E}_{\mathcal{P}_i} \in \Theta} \, e(\mathcal{E}_{\mathcal{P}_i})). \tag{2}$$

## 3 METHODOLOGY

The proposed framework (Figure 2) operates through a *discrete→continuous→discrete* pipeline: (1) **Encoding**: map discrete configurations $\mathcal{P}$ (each $r_i \in \mathcal{C}$) to continuous representations $\mathcal{E}_{\mathcal{P}} \in \mathbb{R}^{k \times d}$; (2) **Optimization**: perform gradient ascent on evaluator $e(\mathcal{E}_{\mathcal{P}})$ in the continuous space $\Theta$; (3) **Decoding**: convert optimized $\mathcal{E}_{\mathcal{P}^*}$ back to implementable discrete $\mathcal{P}^*$ via decoder $g$. Unlike heuristic methods that generate single configurations without further refinement, this continuous relaxation enables iterative gradient-based optimization from multiple promising starting points. The framework comprises four stages: (1) acquisition of "pruning-score" data pairs comprising pruning ratios and corresponding comprehensive scores to serve as training data; (2) development of an encoder-evaluator-decoder architecture to learn a continuous representation space $\Theta$ for pruning optimization, leveraging the collected data pairs; (3) gradient-steered optimization within $\Theta$ to identify the optimal pruning representation; (4) generation of the optimal pruning configuration utilizing the well-trained decoder, based on the optimal pruning representation obtained in the preceding step.

### 3.1 "PRUNING-SCORE" PAIR DATA COLLECTION

Collecting a diverse and high-quality dataset $\mathcal{R} = \{(\mathcal{P}_i, s_i)\}_1^n$ of "pruning-score" pairs is crucial for training effective models. $\mathcal{R}$ should be comprehensive, representing the entire distribution, and include high-performing pruning cases alongside random exploration samples. Classical LLM pruning algorithms enable automated data collection via two complementary strategies. First, we employ exploratory sampling methods (Random-Layer and Random-Block) to generate 50 diverse configurations, ensuring broad coverage of the configuration space. Second, we leverage 10 knowledge-driven methods to generate 50 high-quality configurations: 3 custom heuristics (PPL-based, Energy-based, and Score-based) and 7 established baseline methods (L2, ShortGPT, SliceGPT, Channel, LLM-Pruner, Wanda, and LoRA-Prune). To enhance training robustness, we apply 10× Mixup augmentation (Zhang et al., 2018) to expand the 100 base configurations into 1,000 pruning-score pairs used for training. Complete details are provided in Appendix C. These manually-engineered algorithms reflect varying expert considerations, producing comprehensive "pruning-score" pairs reflecting diverse

expert knowledge. The holistic metric $\mathcal{F}$ facilitates obtaining scores $s_i$ for each collected pruning sample $\mathcal{P}_i$, completing the paired dataset collection. While `Alps` bootstraps from heuristic-based pruning, its learned representation enables novel, optimized policies beyond the input pool via gradient exploration, demonstrating generalization rather than memorization.

## 3.2 CONTINUOUS SPACE FOR PRUNING OPTIMIZATION

**Bridging Discrete and Continuous Spaces.** The configuration space contains $|\mathcal{C}|^k$ discrete combinations (e.g., $101^{32} \approx 10^{64}$ for 32-layer Llama-7B) where gradient-based methods are inapplicable. We address this by learning a continuous representation space $\Theta \subset \mathbb{R}^{k \times d}$ where: (1) each point $\mathcal{E}_{\mathcal{P}}$ encodes a configuration; (2) the evaluator $e(\mathcal{E}_{\mathcal{P}})$ is differentiable, enabling gradient ascent to discover improved representations; (3) the decoder maps optimized representations back to implementable configurations. This transforms intractable discrete search into efficient continuous optimization.

In Section 3.1, we gather a dataset $\mathcal{R}$ of "pruning-score" pairs from the extensive experience of classical LLM pruning techniques. This data is used to train an encoder-evaluator-decoder framework that embeds the pruning configurations into a continuous representation space $\Theta$, enabling gradient-steered optimization of selection. Each point in $\Theta$ corresponds to a pruning configuration $\mathcal{P}$ with an associated comprehensive score $s$. Each pruning configuration $\mathcal{P} = [r_1, r_2, \ldots r_k]$ is naturally a sequence of the pruning ratios for the $k$ layers of a LLM. Thus, the encoder and decoder together in our framework essentially belong to the sequence-to-sequence (seq2seq) architecture (Sutskever et al., 2014), comprising an encoder module to encode the input context and a decoder module to generate the target output sequence based on the encoded representation.

**Encoder $q$.** For a given discrete pruning configuration $\mathcal{P} = [r_1, r_2, \ldots r_k]$ (each $r_i \in \mathcal{C}$), the encoder $q$ maps $\mathcal{P}$ to a continuous representation $\mathcal{E}_{\mathcal{P}} = q(\mathcal{P}) \in \mathbb{R}^{k \times d}$ serving as the input context. we adopt a single-layer LSTM (Hochreiter, 1997) due to its simplicity and efficiency in modeling short pruning configuration sequences; however, our framework is agnostic to the encoder/decoder backbone and could easily support Transformer-based modules. Specifically, pruning ratios $r_i$ from $\mathcal{P}$ are sequentially input, with the corresponding output embeddings $[h_1^e, h_2^e, \ldots, h_k^e] \in \mathbb{R}^{k \times d}$ forming $\mathcal{E}_{\mathcal{P}}$, where $k$ denotes the number of LLMs layers and $d$ the embedding dimension.

**Decoder $g$.** The decoder $g$ aims to generate the pruning ratios of a pruning configuration $\mathcal{P}$ based on the encoded representation $\mathcal{E}_{\mathcal{P}}$, denoted by $\mathcal{P} = g(\mathcal{E}_{\mathcal{P}})$. Analogous to the encoder, a single-layer LSTM is employed to implement the decoder, and trained autoregressively (Sutskever et al., 2014; Brown et al., 2020). Firstly, the decoder's initial input is the final pruning ratio embedding in $\mathcal{E}_{\mathcal{P}}$ (i.e., $h_k^e$). Subsequently, at decoding step $i$, the current decoder hidden state $h_i^d$ is obtained from the decoder LSTM, and dot product attention (Luong et al., 2015b) is utilized to aggregate the input context $\mathcal{E}_{\mathcal{P}}$ from the encoder LSTM, yielding an enhanced input embedding $h_i^{en}$ for the current step, formulated as:

$$h_i^{en} = \sum_{h_j^e \in \mathcal{E}_{\mathcal{P}}} a_{ij} h_j^e, \text{ where } a_{ij} = \frac{\exp\left(h_i^d \cdot h_j^e\right)}{\sum_{h_k^e \in \mathcal{E}_{\mathcal{P}}} \exp\left(h_i^d \cdot h_k^e\right)}, \tag{3}$$

where $a_{ij}$ denotes the attention weight between the decoder hidden state $h_i^d$ and the encoder hidden state $h_j^e$. Subsequently, $h_i^d$ and the attended context vector $h_i^{en}$ are concatenated and passed through a fully connected layer followed by a softmax, yielding the predictive distribution for step $i$, as:

$$P_g\left(r_i \mid \mathcal{P}_{<i}, \mathcal{E}_{\mathcal{P}}\right) = \frac{\exp\left(W_{r_i}\left[h_i^d; h_i^{en}\right]\right)}{\sum_{c \in \mathcal{C}} \exp\left(W_c\left[h_i^e; h_i^{en}\right]\right)}, \tag{4}$$

here, $r_i$ is the pruning ratio for layer $i$, and $\mathcal{P}_{<i}$ are the previous ratios. The set $\mathcal{C}$ forms the core of our decoding strategy: it serves as a vocabulary of 101 discrete pruning ratios from 0.00 to 1.00, making the decoding process analogous to token generation in sequence-to-sequence models. Finally, $W$ parameterizes the fully connected layer. By multiplying the probabilities across steps, the distribution over the entire pruning configuration $\mathcal{P}$ is derived as:

$$P_g(\mathcal{P} \mid \mathcal{E}_{\mathcal{P}}) = \prod_{t=1}^{k} P_g\left(r_i \mid \mathcal{P}_{<i}, \mathcal{E}_{\mathcal{P}}\right). \tag{5}$$

To encourage the generated sequence to resemble the ground truth, the negative log-likelihood of the distribution is minimized, formulated as:

$$\mathcal{L}_{seq2seq} = -\log P_g(\mathcal{P} \mid \mathcal{E}_{\mathcal{P}}) = -\sum_{t=1}^{k} \log P_g\left(r_i \mid \mathcal{P}_{<i}, \mathcal{E}_{\mathcal{P}}\right). \tag{6}$$

**Evaluator** $e$. The evaluator $e$ aims to evaluate the comprehensive score $s$ of a pruning configuration $\mathcal{P}$ based on its encoded representation $\mathcal{E}_{\mathcal{P}}$. Specifically, mean pooling is applied to the pruning ratio embeddings $h^e_{(.)}$ in $\mathcal{E}_{\mathcal{P}}$ to obtain the integrated pruning embedding $\bar{\mathcal{E}}_{\mathcal{P}} \in \mathbb{R}^d$, which aggregates the pruning information. $\bar{\mathcal{E}}_{\mathcal{P}}$ is then input to the evaluator $e$ (a feedforward neural network) to estimate the score $\hat{s} = e(\bar{\mathcal{E}}_{\mathcal{P}})$. To minimize the discrepancy between the estimated $\hat{s}$ and the collected ground truth $s$, the Mean Squared Error (MSE) loss is employed, formulated as:

$$\mathcal{L}_{score} = \text{MSE}(s, \hat{s}) = (s - \hat{s})^2. \tag{7}$$

**Joint Training Loss** $\mathcal{L}$. The encoder $q$, decoder $g$, and evaluator $e$ are optimized jointly by integrating the seq2seq loss (Equation 6) and the score estimation loss (Equation 7) into a joint training objective $\mathcal{L}$, formulated as:

$$\mathcal{L} = \alpha \mathcal{L}_{seq2seq} + (1 - \alpha)\mathcal{L}_{score}, \tag{8}$$

here, $\alpha$ is a hyperparameter that balances the contributions of the seq2seq loss $\mathcal{L}_{seq2seq}$ and the score estimation loss $\mathcal{L}_{score}$ during the joint training procedure.

### 3.3 GRADIENT-STEERED PRUNING OPTIMIZATION

With the trained encoder, evaluator, and decoder, a gradient-steered optimization method is employed in $\Theta$ to identify the optimal pruning configuration. Good initialization plays a pivotal role in the gradient-steered optimization approaches (Glorot & Bengio, 2010), thus the top-$K$ collected pruning configurations ranked by their score $s$ are embedded via the encoder to continuous representations, serving as starting points for subsequent optimization. Denoting one such starting point representation as $\mathcal{E}_{\mathcal{P}}$, to obtain a configuration with an improved comprehensive score, optimization proceeds from $\mathcal{E}_{\mathcal{P}}$ along the gradient direction induced by the evaluator $e$:

$$\mathcal{E}_{\mathcal{P}}^+ = \mathcal{E}_{\mathcal{P}} + \eta \frac{\partial e(\mathcal{E}_{\mathcal{P}})}{\partial \mathcal{E}_{\mathcal{P}}}, \tag{9}$$

here, $\mathcal{E}_{\mathcal{P}}^+$ denotes the optimized pruning configuration representation, $\eta$ is the step size, and $e(\mathcal{E}_{\mathcal{P}}^+) \geq e(\mathcal{E}_{\mathcal{P}})$ is obvious that implies an improved comprehensive score for $\mathcal{E}_{\mathcal{P}}^+$ compared to the initial $\mathcal{E}_{\mathcal{P}}$. This optimization is repeated for all $K$ starting points, yielding a set of candidate pruning representations $\{\tilde{\mathcal{E}}_{\mathcal{P}_i}\}_1^K$. The optimal pruning representation $\mathcal{E}_{\mathcal{P}^*}$ is finally selected from $\{\tilde{\mathcal{E}}_{\mathcal{P}_i}\}_1^K$ based on the estimated comprehensive scores, *i.e.*, $\mathcal{E}_{\mathcal{P}^*} = \text{argmax}_{\tilde{\mathcal{E}}_{\mathcal{P}_i}} \{e(\tilde{\mathcal{E}}_{\mathcal{P}_i})\}_1^K$.

### 3.4 OPTIMAL PRUNING GENERATION

The final step is to decode the optimal continuous representation $\mathcal{E}_{\mathcal{P}^*}$ into an implementable discrete configuration $\mathcal{P}^* = g(\mathcal{E}_{\mathcal{P}^*})$. To achieve this, we employ beam search (Freitag & Al-Onaizan, 2017), a standard technique in NLP for generating sequences (Sutskever et al., 2014). The process is directly analogous to text generation. The decoder treats the set of candidate ratios $\mathcal{C} = \{0.00, 0.01, \ldots, 1.00\}$ as a discrete vocabulary. The generation is constrained to produce a sequence of exactly $k$ such ratios, one for each of the $k$ layers. The final result is the optimal configuration $\mathcal{P}^*$, a fine-grained plan where each generated ratio directly guides the removal of a corresponding proportion of structural blocks within its layer, enabling adaptive and efficient compression.

## 4 EXPERIMENTS

### 4.1 EXPERIMENTAL SETUP

**Design Context.** Alps is designed for production edge deployment where efficiency impacts user experience across device fleets. The framework requires one-time optimization to generate

hardware-aware strategies, trading upfront cost for sustained gains. Existing methods produce **universal** configurations—single solutions for all contexts. In contrast, `Alps` is a **deployment-specific optimizer**: given a device with specific latency/energy budgets, it discovers a Pareto-optimal configuration for that scenario. This paradigm shift—from one-size-fits-all heuristics to per-deployment compilation—justifies the upfront cost, as benefits accumulate across deployment.

**Base Models.** To showcase the effectiveness and versatility `Alps`, we evaluate it over three open-source LLMs: Llama2-7B, Llama2-13B (Touvron et al., 2023), Llama3-8B, Qwen1.5-7b, Qwen2.5-7b (Yang et al., 2025) and Vicuna-7B (Chiang et al., 2023).

**Evaluation Metrics.** In order to comprehensively evaluate the effectiveness and versatility of `Alps`, we conduct evaluations from three perspectives. (1) **Generation Ability.** We assess the model's post-pruning generation capabilities using zero-shot perplexity (PPL) on the WikiText2 and PTB datasets. Lower PPL values signify stronger generative performance, indicating the model's ability to effectively handle language generation tasks even after significant model reductions. (2) **General-purpose Task Solving Ability.** To demonstrate the world knowledge and problem-solving skills of the customized model, higher classification accuracy indicates stronger performance. We utilize two benchmark: *(a) BBH* (Srivastava et al., 2022), which includes 23 challenging tasks such as Q&A, natural language reasoning, and sentiment analysis. *(b)* Commonsense reasoning tasks used in Llama paper (Touvron et al., 2023), including BoolQ (Clark et al., 2019), PIQA (Bisk et al., 2020), OpenbookQA (Mihaylov et al., 2018), ARC-challenge (Clark et al., 2018), ARC-easy (Clark et al., 2018), WinoGrande (Sakaguchi et al., 2021), and HellaSwag (Zellers et al., 2019). (3) **System Effectiveness.** In the context of resource-constrained edge devices, efficient and low-cost operation is crucial for deployment. We leverage Codecarbon (cod, 2023) to track latency and energy consumption during the inference on WikiText2 (Merity et al., 2017).

**Baselines.** We compare `Alps` with 13 representative pruning methods: **Layer-wise:** *(1) Random-Layer*: random layer removal; *(2) PPL-Layer*: removes layers by perplexity ranking; *(3) Short-GPT* (Men et al., 2024): layer pruning via Block Influence metric. **Block-wise:** *(4) Random-Block*: random group pruning; *(5) L2*: magnitude-based group importance; *(6) SliceGPT* (Ashkboos et al., 2024): matrix factorization to reduce embedding dimensions; *(7) Channel* (Ma et al., 2023): gradient-based channel-wise pruning; *(8) LLMPruner* (Ma et al., 2023): gradient-based attention head pruning. **Weight-level:** *(9) Wanda* (Sun et al., 2023): magnitude and activation-aware unstructured pruning; *(10) LoRA-Prune* (Zhang et al., 2023): structured sparsity with low-rank adaptation; (11) AdaPruner (Kong et al., 2025): uses Bayesian optimization to automatically select the best calibration data and metrics for effective structural pruning; (12) MoDeGPT Lin et al. (2025): reduces hidden dimensions by applying modular decomposition techniques; (13) SVD-LLM Wang et al. (2024): utilizes truncation-aware data whitening to ensure singular values accurately map to model compression loss, enabling effective high-ratio pruning.

**Hyperparameter Settings and Reproducibility.** We execute heuristic approaches for 100 configurations to collect training data. We then applied a $10\times$ Mixup data augmentation to expand it for training. The Encoder and Decoder utilize an identical architecture, specifically a single-layer LSTM configuration. The Evaluator employs a dual-layer feed-forward architecture. The dimensionality of the hidden states for the Encoder, Decoder, and Evaluator is set at 64, 64, and 200 respectively. The embedding size of each model layer is 32. For `Alps` training, we configure the batch size as 1024, the learning rate as 0.001, and $\alpha$ as 0.8 respectively. For optimization, we start with the top 25 model pruning ratio records. See Appendix B for more details.

## 4.2 MAIN PERFORMANCE

**Model Performance.** To evaluate the efficacy of `Alps` in generating efficient pruning configurations that explore the optimal pruning space while ensuring high performance, we conducted comprehensive experiments. Table 1 details the zero-shot performance of the pruned model across a variety of downstream tasks, as well as its zero-shot perplexity on the WikiText2 and PTB datasets, utilizing diverse pruning configurations. Figure 3 illustrates the pruned performance on BBH benchmarks. Our observations confirm that `Alps` consistently outperforms established baselines across all metrics. Specifically, when applied to the Llama2-7B model with a 20% pruning ratio, `Alps` retains superior generative capabilities, achieving up to $289.43\times$ the generative capacity of the *Random-Layer* on the WikiText2 and up to $59.26\times$ on PTB. This remarkable performance can be attributed to `Alps`'s

Table 1: Zero-shot performance (7 commonsense reasoning datasets) and perplexity (2 datasets) of the 3 pruned models with 2 pruning ratio settings over different pruning schemes. **Bold** and Underlined indicate the highest and second highest scores, respectively. Additional results on LLaMA2-13B and Vicuna-7B at 50% sparsity, and Llama-7B, Llama3-8B, Qwen1.5-7B, Qwen2.5-7B at both 20% and 50%, are provided in Appendix D.

| Method | Perplexity ↓ | | | Downstream task performance (%) ↑ | | | | | | | |
|---|---|---|---|---|---|---|---|---|---|---|---|
| | WikiText2 | PTB | Average | BoolQ | PIQA | OBQA | ARC-c | ARC-e | WinoG | HellaSwag | Average |
| Llama2-7B | 12.18 | 48.37 | 30.28 | 77.40 | 78.80 | 58.60 | 45.90 | 75.20 | 69.20 | 77.20 | 68.90 |
| Llama2-7B (20%) | | | | | | | | | | | |
| Random-Layer | 4726.49 | 3623.92 | 4175.21 | 43.76 | 53.10 | 27.00 | 28.84 | 26.60 | 51.30 | 26.57 | 36.74 |
| PPL-Layer | 24.28 | 73.47 | 48.87 | 59.82 | 71.11 | 36.40 | 35.07 | 55.13 | 59.83 | 62.12 | 54.21 |
| Random-Block | 82.92 | 209.29 | 146.10 | 60.28 | 68.12 | 31.6 | 30.03 | 52.23 | 49.88 | 39.41 | 47.36 |
| L2 | 821.34 | 2432.98 | 1627.16 | 50.67 | 58.38 | 31.00 | 27.99 | 31.94 | 49.01 | 33.2 | 40.31 |
| ShortGPT | 29.16 | 84.89 | 57.03 | 62.14 | 70.51 | 35.40 | 33.45 | 48.15 | 61.56 | 59.44 | 52.95 |
| SliceGPT | 28.65 | 154.18 | 91.41 | **72.69** | 70.73 | 35.20 | 38.82 | 59.09 | 62.27 | 60.60 | 57.06 |
| Channel | 20.32 | 82.28 | 51.30 | 67.83 | 73.45 | **39.40** | 34.30 | 61.45 | 61.40 | 63.48 | 57.33 |
| LLMPruner | 17.35 | 81.01 | 45.41 | 62.20 | **77.64** | 39.40 | 38.31 | 65.24 | 62.75 | 68.24 | 59.11 |
| Wanda | 19.78 | 70.18 | 44.98 | 66.81 | 76.44 | 37.80 | 35.10 | 61.38 | 63.92 | 66.78 | 58.32 |
| LoraPrune | 17.84 | 66.32 | 42.08 | 67.05 | 76.92 | 38.10 | 37.22 | 63.34 | 64.52 | 68.39 | 59.36 |
| SVD-LLM | 19.20 | 71.50 | 45.35 | 66.80 | 75.20 | 37.80 | 36.10 | 62.50 | 63.50 | 67.10 | 58.43 |
| MoDeGPT | 17.15 | 64.20 | 40.68 | 70.20 | 77.10 | 38.50 | 38.40 | 66.50 | 65.50 | 68.90 | 60.73 |
| AdaPruner | 17.50 | 65.80 | 41.65 | 69.50 | 76.80 | 38.20 | 38.10 | 65.90 | 65.80 | 69.10 | 60.49 |
| **ALPS** | **16.33** | **61.15** | **38.73** | 71.13 | 77.37 | 39.00 | **39.08** | **68.27** | **67.64** | **69.81** | **61.76** |
| Llama2-7B (50%) | | | | | | | | | | | |
| Random-Layer | 10903.89 | 9326.59 | 10115.24 | 38.10 | 53.10 | 28.20 | 27.99 | 26.18 | 50.43 | 25.81 | 35.69 |
| PPL-Layer | 183.27 | 573.39 | 378.33 | 48.56 | 56.26 | 31.40 | 27.39 | 31.31 | 51.38 | 31.21 | 39.64 |
| Random-Block | 557.92 | 498.17 | 528.04 | 60.06 | 61.53 | 29.40 | 26.88 | 40.91 | 50.43 | 38.19 | 43.91 |
| L2 | 10819.04 | 1071.23 | 5945.14 | 59.66 | 60.5 | 28.60 | 27.05 | 30.13 | 49.96 | 27.03 | 40.42 |
| ShortGPT | 1054.62 | 1894.81 | 1474.72 | 60.06 | 57.29 | 32.00 | 29.10 | 29.55 | **53.20** | 37.40 | 42.66 |
| SliceGPT | 224.86 | 946.98 | 585.92 | **62.26** | 58.81 | 27.80 | 26.71 | 41.62 | 52.09 | 36.87 | 43.74 |
| Channel | 115.13 | 477.22 | 296.17 | 61.35 | 59.03 | 29.20 | 25.17 | 35.77 | 52.25 | 33.16 | 42.28 |
| LLMPruner | 51.39 | 262.52 | 156.95 | 49.30 | 65.18 | 33.80 | 27.47 | 43.18 | 51.22 | 39.01 | 44.17 |
| Wanda | 41.56 | 184.51 | 113.04 | 48.95 | 60.34 | 30.60 | 28.84 | 45.21 | 50.56 | 40.34 | 43.55 |
| LoraPrune | 44.89 | 177.36 | 111.13 | 50.07 | 61.12 | 31.30 | **29.65** | 46.32 | 51.78 | 42.91 | 44.74 |
| SVD-LLM | 48.50 | 195.20 | 121.85 | 50.10 | 60.50 | 31.50 | 27.80 | 45.10 | 50.80 | 41.20 | 43.86 |
| MoDeGPT | 39.80 | 150.50 | 95.15 | 55.50 | 64.80 | 34.20 | 28.80 | 47.20 | 51.50 | 43.50 | 46.50 |
| AdaPruner | 40.20 | 155.80 | 98.00 | 54.80 | 64.20 | 34.50 | 28.50 | 47.50 | 51.20 | 43.10 | 46.26 |
| **Alps** | **35.66** | **112.03** | **73.85** | 51.87 | **67.30** | **35.40** | 29.35 | **50.55** | 52.25 | **45.43** | **47.45** |
| Vicuna-7B | 16.23 | 60.67 | 38.45 | 75.72 | 77.09 | 42.40 | 39.85 | 69.07 | 67.80 | 71.06 | 63.28 |
| Vicuna-7B (20%) | | | | | | | | | | | |
| Random-Layer | 5882.21 | 3887.90 | 4885.06 | 53.58 | 53.92 | 28.20 | 27.82 | 25.84 | 50.59 | 26.91 | 38.12 |
| PPL-Layer | 30.87 | 92.87 | 61.87 | **69.17** | 70.02 | 37.60 | 35.24 | 56.31 | 61.01 | 59.96 | 55.62 |
| Random-Block | 97.71 | 204.45 | 151.08 | 65.08 | 67.95 | 33.00 | 30.20 | 56.52 | 52.17 | 43.92 | 49.83 |
| L2 | 1504.79 | 2180.92 | 1842.85 | 52.69 | 56.15 | 33.20 | 27.39 | 34.01 | 52.09 | 32.63 | 41.17 |
| ShortGPT | 36.02 | 105.24 | 70.63 | 62.57 | 68.99 | 37.40 | 35.07 | 56.90 | 62.04 | 61.59 | 54.94 |
| SliceGPT | 21.82 | 148.02 | 84.92 | 53.70 | 64.47 | 33.00 | 34.30 | 52.31 | 60.22 | 49.26 | 49.61 |
| Channel | 90.72 | 305.72 | 198.22 | 64.37 | 63.98 | 33.60 | 28.33 | 49.83 | 53.91 | 44.92 | 48.42 |
| LLMPruner | 22.98 | 80.69 | 51.84 | 64.77 | 75.35 | 38.00 | **38.99** | **67.30** | 63.69 | 64.62 | 58.96 |
| Wanda | 23.41 | 78.34 | 50.88 | 64.01 | 73.44 | 36.40 | 34.24 | 63.44 | 62.92 | 65.81 | 57.18 |
| LoraPrune | 21.86 | 85.26 | 53.56 | 60.12 | 70.02 | 35.90 | 34.22 | 58.37 | 56.81 | 60.43 | 53.70 |
| **Alps** | **21.76** | **73.18** | **47.47** | 67.86 | **75.84** | **41.20** | 38.57 | 66.37 | **63.93** | **67.74** | **60.22** |
| Llama2-13B | 10.98 | 54.39 | 32.69 | 81.70 | 80.50 | 57.00 | 49.40 | 77.30 | 72.80 | 80.70 | 71.34 |
| Llama2-13B (20%) | | | | | | | | | | | |
| Random-Layer | 47.62 | 310.53 | 179.08 | 61.28 | 63.76 | 33.20 | 30.03 | 50.34 | 56.27 | 42.45 | 48.19 |
| PPL-Layer | 15.13 | 69.83 | 42.48 | 62.23 | 75.08 | 40.20 | 38.23 | 62.63 | 66.93 | 70.31 | 59.37 |
| Random-Block | 43.96 | 146.11 | 95.04 | 62.17 | 73.61 | 34.80 | 36.18 | 64.06 | 56.99 | 60.04 | 55.40 |
| L2 | 634.68 | 1083.86 | 859.27 | 49.08 | 59.90 | 27.00 | 25.51 | 33.92 | 48.7 | 35.36 | 39.92 |
| ShortGPT | 16.52 | 72.61 | 44.57 | 62.91 | 74.16 | 40.80 | 40.27 | 62.88 | 67.25 | 69.90 | 59.74 |
| SliceGPT | 23.33 | 125.57 | 74.45 | 74.34 | 72.74 | **43.40** | 41.55 | 65.11 | 68.51 | 64.18 | 61.40 |
| Channel | 16.68 | 80.69 | 48.69 | **76.42** | 77.37 | 39.00 | 39.85 | 69.11 | 63.61 | 69.57 | 62.13 |
| LLMPruner | 16.29 | 83.25 | 49.77 | 72.78 | **81.00** | 42.40 | 42.92 | 72.10 | 67.25 | 73.94 | 64.63 |
| Wanda | 15.66 | 71.40 | 43.53 | 71.05 | 77.82 | 41.80 | 39.43 | 70.44 | 67.34 | 72.61 | 62.93 |
| LoraPrune | 16.19 | 69.23 | 42.71 | 72.44 | 78.37 | 42.40 | 40.92 | 71.38 | **68.21** | 74.16 | 63.98 |
| **Alps** | **13.75** | 67.41 | **40.58** | 72.51 | 79.73 | 42.60 | **43.60** | **73.53** | 67.88 | **75.25** | **65.01** |

utilization of Perplexity as a pruning metric, which enables the construction of a continuous pruning selection space. Furthermore, Alps significantly enhances the average accuracy, surpassing the Random-Layer by 68.10% and 17.82% than other baselines on average. In the BBH tests, Alps records a 13.75% improvement over Random-Block and a 4.1% average increase over other baselines. The pruning configurations generated by Alpsre not simple binary masks or fixed-ratio schemes. Instead, Alpsroduces structured pruning vectors of continuous values in [0,1], specifying per-layer block-wise sparsity. This allows differentiated pruning per layer, enabling fine-grained adaptation to layer sensitivity and system constraints. As detailed in Section 3, the driver is that Alps utilizes gradient-steered optimization to precisely identify the optimal pruning configurations. Furthermore,

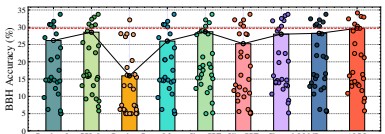

Figure 3: 5-shot BBH performance.

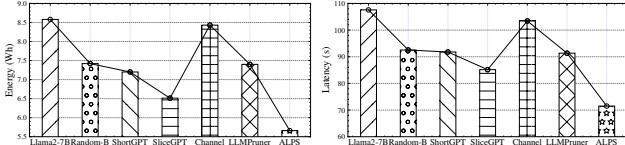

Figure 4: Energy and latency comparison.

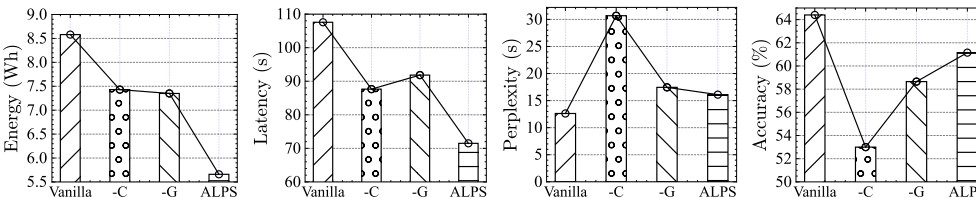

Figure 5: Ablation study. The influence of energy consumption ($\downarrow$), latency ($\downarrow$), perplexity ($\downarrow$), and task performance ($\uparrow$) of data collection ($\texttt{Alps}^{-C}$), and Gradient Optimization ($\texttt{Alps}^{-G}$) with inference pruned model on WikiText2.

we conducted more advanced experiments to analyze $\texttt{Alps}$ in: 1) ***Different model pruning ratios***: As shown in Table 1, for Llama2-7B at pruning ratios of 20% and 50%, we observe that even at a substantial 50% pruning ratio, there is a performance improvement of 14.68% over the baseline models on average, effectively preserving the functionality. 2) ***Different model types***: Comparing Llama2-7B and Vicuna-7B, the results indicate that $\texttt{Alps}$ exhibits a strong generalization capability, outperforming baselines by an average of 23.85%. 3) ***Different model sizes***: Evaluating Llama2-7B and Llama2-13B, it is evident from Table 1 that $\texttt{Alps}$ maintains 91.13% of the performance of the unpruned models, underscoring its scalability. These findings, with additional experiments (other model & pruning ratio) provided in Appendix D, collectively demonstrate the practical effectiveness and robust scalability of $\texttt{Alps}$ across complex workloads and diverse application scenarios.

**System Effectiveness.** We evaluate the system effectiveness of $\texttt{Alps}$ from two critical perspectives: inference energy consumption and latency. The experiments, conducted on a single NVIDIA A40 graphics card and using the WikiText2 test set to inference, are closely monitored using Codecarbon (cod, 2023). This setup allows us to measure both the energy consumption during inference and the time duration of the process. As shown in Figure 4, $\texttt{Alps}$ demonstrates significant improvements in efficiency, achieving energy savings of up to 34.1% and reducing inference latency by as much as 33.5%. These benefits are the result of $\texttt{Alps}$'s integrated approach to training, where it parallelizes and optimizes for both latency and energy consumption, closely tracking the needs of real-world applications, particularly in mobile and other battery-powered environments. In contrast, other pruning approaches, such as LLMPruner, which focuses primarily on block performance importance, and ShortGPT, which relies on hidden state analysis to determine layer influence, do not incorporate energy and latency considerations into their pruning metrics. This oversight leads to suboptimal performance in practical scenarios, highlighting the advanced and holistic manner of $\texttt{Alps}$.

### 4.3 METHOD ANALYSIS

**Ablation Study.** To evaluate the importance and impact of different components within $\texttt{Alps}$, we developed two experimental model variants for comparative analysis: $\texttt{Alps}^{-C}$, which randomly collects pairwise data without relying on classical model pruning methods. $\texttt{Alps}^{-G}$, where the gradient optimization process of $\texttt{Alps}$ is disabled, and the optimal pruning configurations are selected from the existing pruning score data. Figure 5 shows the results of the comparison, demonstrating that $\texttt{Alps}$ outperforms $\texttt{Alps}^{-C}$. Specifically, $\texttt{Alps}$ shows a $1.91\times$ improvement in generation capability and an 8.15% increase in task performance compared to $\texttt{Alps}^{-C}$. It also achieves a $1.22\times$ reduction in inference latency and a 23.82% reduction in power consumption. Gradient-steered optimization mainly enhances pruning utility under constraints, hence showing stronger gains in system metrics. These improvements underscore the effectiveness of using classically derived pruning sets, which provide robustness and superior denoising capabilities critical for optimizing the representation space and selecting effective pruning configurations. Furthermore, $\texttt{Alps}$ outperforms $\texttt{Alps}^{-G}$, improving generation capability by $1.28\times$, downstream task performance by 2.50%, inference latency by $1.10\times$, and reducing energy cost by 22.9%. The lackluster performance of $\texttt{Alps}^{-G}$, particularly in energy and latency, illustrates the importance of integrating system metrics

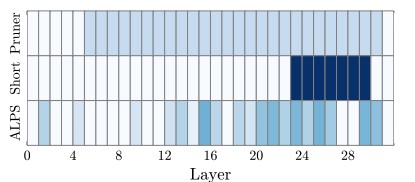

Figure 6: Layer-wise pruning ratio.

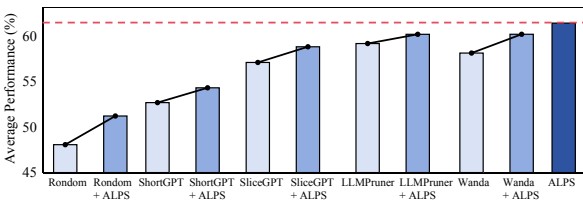

Figure 7: Generalization ability of Alps.

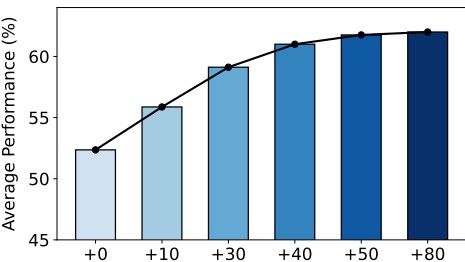

Figure 8: Sensitivity analysis of heuristic points. Fixing the random samples at 50, we varied the number of heuristic samples across $\{0, 10, 30, 50, 80\}$.

into pruning decisions traditional approaches overlook. The gradient optimization component is critical, enhancing data diversity and accurately modeling the order-independent attributes of model pruning, thus promoting more effective and robust learning. Consequently, these results highlight that data collection and gradient optimization are essential for maintaining Alps performance.

**Study of Pruning Strategy Made by Alps.** To compare pruning strategies, we analyze layer-wise pruning ratios under a fixed 20% overall budget for Alps, ShortGPT (Men et al., 2024), and LLMPruner (Ma et al., 2023), as shown in Figure 6. The visualization reveals distinct patterns: LLMPruner applies uniform mid-range pruning across most layers, showing limited adaptability; ShortGPT implements a binary approach that completely removes certain consecutive layers while preserving others. In contrast, Alps generates a diverse, non-uniform distribution of pruning ratios—automatically derived through gradient optimization rather than manually specified. This layer-adaptive approach selectively preserves critical layers while applying varied pruning intensities to others, enabling better performance-efficiency trade-offs.

**Study of Generalization Ability of Alps.** Alps is a feature for creating a diverse knowledge base. The goal is not to imitate a single heuristic, but to learn a generalized representation of "what makes a good pruning configuration" from many different "experts". Figure 7 outlines when baselining the Llama2-7B (20% pruning ratio), the generalizability of Alps in combination with different pruning methods and the effectiveness of using multiple pruning ratio for data collection . The adoption of a single pruning method as data collector can significantly improve results over the corresponding baseline, highlighting the merits of continuous space paradigms in enhancing generalizability. Furthermore, the use of diverse, multiple pruning methods as data collector (*i.e.*, the Alps) outperforms the single strategy, emphasizing the value of diversity in data collector for comprehensive space construction.

**Sensitivity Analysis of Heuristic Configurations.** We evaluated the sensitivity of our approach to the number of heuristic configurations ($\{0, 10, 30, 50, 80\}$) given 50 fixed random samples. Results show that heuristic initialization provides a crucial warm-start, with performance peaking around 30–50 samples. The observed saturation indicates that key redundancy patterns are learned quickly, after which additional samples provide negligible information gain. Coupled with Mixup augmentation to densify the feature space, a set of 50 heuristic points proves sufficient. We therefore adopt this setting as a cost-effective equilibrium between search efficiency and performance.

## 5 RELATED WORK

While traditional CNN/DNN pruning methods (Yang et al., 2017; He et al., 2017) harm the generative capabilities of LLMs, specialized techniques (Ma et al., 2023; Ashkboos et al., 2024; Men et al., 2024; Yang et al., 2024) have been developed. However, these approaches are often in heuristic manner and neglect system-level metrics like energy consumption and inference latency, which are critical for deployment on resource-constrained devices. For instance, SliceGPT (Ashkboos et al., 2024) employs matrix factorization of hidden states for structured compression, while LLM-Pruner (Ma et al., 2023) removes non-essential structures using gradient-steered criteria. Plug-and-play `Alps` converts discrete to continuous spaces using a unique generator-decoder architecture to dynamically generate optimal pruning configurations. While unstructured pruning (Zhang et al., 2024; Li et al., 2024) offers parameter-level sparsity, `Alps` focuses on structured block removal for real-world deployment compatibility. `Alps` complements structured approaches like BlockPruner(Zhong et al., 2024) and SLEB (Song et al., 2024), but extends them with adaptive search. Existing customization methods (Li et al., 2020b;a) fail for task-agnostic LLMs, as their auto-regressive nature poses unique pruning challenges. `Alps` addresses this by using an LSTM-based architecture to embed pruning ratios into a continuous space, which fits the sequential structure of LLM layers. `Alps` then apply gradient ascent to this representation to find an optimal pruning policy.

## 6 CONCLUSION

LLMs offer remarkable capabilities but their billion-parameter scale creates significant deployment challenges on edge devices. Current pruning approaches rely on discrete-space heuristics, often neglecting system efficiency metrics crucial for real-world deployment. We introduced `Alps`, which reformulates the search for optimal pruning configurations as a gradient-steered optimization in continuous space. Our approach automatically curates training data, learns a continuous representation space, performs gradient optimization, and decodes results into implementable configurations. Extensive experiments across multiple LLM architectures demonstrate `Alps`'s effectiveness in simultaneously reducing energy consumption and latency while maintaining high model performance. Our method shows consistent advantages over state-of-the-art approaches, particularly at higher pruning ratios. By jointly optimizing for performance and efficiency, `Alps` offers a comprehensive solution for LLM deployment on resource-constrained devices. Limitations are discussed in Appendix F.

## ETHICS STATEMENT

We affirm adherence to the ICLR Code of Ethics. This work studies compression methods for large language models and does not involve human subjects, personally identifiable information, or sensitive attributes. All datasets and pretrained weights used are publicly available and were accessed and used in accordance with their licenses and terms of use; no data scraping outside the providers' terms was performed. Potential risks include lowering the computational barrier for deploying more capable models in resource-constrained settings; to mitigate misuse concerns, we evaluate only on standard public benchmarks, refrain from releasing domain-specific models for sensitive applications, and provide documentation to support responsible use. The authors take full responsibility for the integrity and accuracy of the reported results.

## REPRODUCIBILITY STATEMENT

We place strong emphasis on the transparency and reproducibility of our work. To facilitate independent verification, the complete implementation has been provided in the supplementary materials, allowing readers to directly reproduce the reported experiments. In addition, Section 4.1 of the main text outlines the experimental pipeline, including dataset preparation, model configurations, and training procedures. For further clarity, we provide the source code as an attachment. Together, these resources ensure that our results can be reliably replicated and extended in future research.

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

## A   THE USE OF LARGE LANGUAGE MODELS

We used LLMs solely as a writing-assistance tool to polish our paper (grammar, wording, concision, and minor LATEX formatting). The LLM did not contribute to research ideation, problem formulation, method design, experiments, data analysis, results, or conclusions, and it was not used to generate citations or technical content. All suggestions were reviewed and, when adopted, edited by the authors, who take full responsibility for the paper's content; no proprietary data beyond the manuscript text was shared with the tool.

## B   EXPERIMENT DETAILS

### B.1   DATASET DETAILS

To evaluate the model's performance in a task-agnostic setting, we utilized seven widely recognized datasets for common sense reasoning: BoolQ (Clark et al., 2019), PIQA (Bisk et al., 2020), HellaSwag (Zellers et al., 2019), WinoGrande (Sakaguchi et al., 2021), ARC_c (Clark et al., 2018), ARC_e (Clark et al., 2018), and OpenbookQA (Mihaylov et al., 2018). The model's efficacy is assessed based on its capacity to accurately rank answers in multiple-choice settings or generate appropriate responses in open-ended scenarios. Further, we supplemented our evaluation with zero-shot perplexity tests on the WikiText2 (Merity et al., 2017) and PTB (Marcus et al., 1993) datasets, which are benchmarks for assessing the generative capabilities of models. Following are their detailed descriptions:

- **BoolQ** is comprised of a question-answering collection designed for yes/no responses, featuring a total of 15,942 examples. These questions arise naturally in settings that are unprompted and unconstrained. Each entry in the dataset includes a triplet format: a question, a corresponding passage, and an answer, with the title of the page provided as optional additional context.

- **PIQA** is an innovative commonsense QA benchmark designed to evaluate reasoning about naive physics, specifically focusing on interactions with everyday objects in common situations. This dataset emphasizes the concept of affordances—what actions are possible with individual physical objects and what interactions are possible among a group of objects. It challenges participants to reason about both the conventional and unconventional, yet feasible, uses of objects. The dataset comprises 20,000 QA pairs, formatted as either multiple-choice or true/false questions.

- **HellaSwag** is a dataset crafted to explore grounded commonsense inference, comprising 70,000 multiple-choice questions derived from real-world scenarios. Each question is sourced from one of two domains: ActivityNet or WikiHow, and includes four potential answers that predict the subsequent event in the scenario. The correct answer corresponds to the actual subsequent event, while the three incorrect choices are carefully crafted through the adversarial generation and verified by humans to challenge machines without misleading humans.

- **WinoGrande** is a comprehensive dataset containing 44,000 problems, modeled after the original Winograd Schema Challenge (WSC) but enhanced to increase both its size and difficulty. The development of this dataset involved two critical phases: firstly, a meticulously planned crowdsourcing process to gather data, and secondly, the implementation of the innovative AfLite algorithm for systematic bias reduction. This algorithm extends beyond human-detectable word associations to include machine-detectable associations within embeddings, ensuring a more robust dataset.

- **ARC** is designed for multiple-choice question answering, featuring questions sourced from science examinations spanning grades 3 to 9. It is divided into two sections: Easy and Challenge, with the Challenge section comprising the more complex questions that demand reasoning abilities. Typically, each question offers four possible answers, though less than 1% of the questions present three or five options. Additionally, ARC is supported by a knowledge base containing 14.3 million unstructured text passages.

- **OpenbookQA** is a novel question-answering dataset inspired by open book exams, aimed at evaluating a person's understanding of a subject. This dataset includes 5,957 multiple-choice

questions at the elementary science level, broken down into 4,957 for training, 500 for development, and 500 for testing. These questions are designed to test comprehension of a concise "book" consisting of 1,326 fundamental science facts and their application to new scenarios.

- **WikiText2** is a corpus composed of more than 100 million tokens sourced from Wikipedia's collection of Good and Featured articles. It offers a substantial increase in size compared to the preprocessed version of the Penn Treebank (PTB), being over twice the size of WikiText-2 and more than 110 times larger for WikiText-103. Unlike PTB, which eliminates original casing, punctuation, and numbers, the WikiText dataset preserves these elements, thereby providing a richer vocabulary. Comprising complete articles, this dataset is particularly beneficial for models designed to leverage long-term dependencies.

- **PTB** is a widely recognized and utilized corpus for evaluating sequence labeling models. The primary task involves tagging each word with its corresponding Part-of-Speech. The standard division of the corpus allocates sections 0 to 18 for training, which includes 38,219 sentences and 912,344 tokens. Sections 19 to 21 are designated for validation, encompassing 5,527 sentences and 131,768 tokens, while sections 22 to 24 are set aside for testing, with 5,462 sentences and 129,654 tokens. PTB is also frequently used in both character-level and word-level language modeling.

## B.2 BASELINES DETAILS

We compare `Alps` with the representative model compression techniques, including:

- **LLM-Pruner** employs a structured pruning technique to efficiently compress LLMs in a task-agnostic manner, reducing reliance on the original training corpus while maintaining the linguistic capabilities of the LLMs. To ensure minimal impact during compression, the LLM-Pruner identifies all interconnected structures within the LLMs and groups them according to their importance, thereby enabling prioritized removal. For the purpose of our experiments, we consider a dataset $\mathcal{D} = \{x_i, y_i\}_{i=1}^N$, where N represents the number of samples, set to 10 in this study, sourced from various public datasets. A group, defined as a set of interconnected structures, is denoted by $\mathcal{G} = \{W_i\}_{i=1}^M$, where M is the number of interconnected structures within the group and $W_i$ represents the weight assigned to each structure. The primary objective during pruning is to eliminate the group that exerts the least influence on the model's predictive performance, typically reflected by changes in the loss function. Specifically, the importance of each weight $W_i$, and the importance of coupled structure can be defined as:

$$I_{W_i^k} = |\frac{\partial \mathcal{L}(\mathcal{D})}{\partial W_i^k} W_i^k - \frac{1}{2} W_i^k H_{kk} W_i^k + \mathcal{O}\left(\|W_i^k\|^3\right)| \qquad (10)$$

Where $k$ represents the k-th parameter in $W_i$. During the experiment, we set the is 1. The $H$ is the hessian matrix. The $\mathcal{L}$ represents the next-token prediction loss. Once the importance of each group is evaluated, we rank these groups accordingly. Groups deemed less critical are then pruned according to a predetermined pruning configuration to optimize the model's efficiency.

- **ShortGPT** employs a layer-wise pruning strategy designed to eliminate redundancy within the layers of LLMs. To identify and remove these redundant layers, a new metric called Block Influence (BI) has been introduced. This metric operates under the assumption that the greater the impact a transformer block has on altering the hidden states, the more crucial that layer is considered. The BI score for the $i^{th}$ block is calculated using the following methodology:

$$\text{BI}_i = 1 - \mathbb{E}_{X,t} \frac{X_{i,t}^T X_{i+1,t}}{\|X_{i,t}\|_2 \|X_{i+1,t}\|_2}, \qquad (11)$$

where $X_{i,t}$ represents the $t^{th}$ row of $X_i$. The process begins by gathering the hidden states from each layer while the model processes the samples during inference. Subsequently, the BI score is computed based on these collected hidden states. Finally, the layers are ranked in ascending order based on their BI scores, and those with lesser importance are pruned from the model.

- **SliceGPT** utilizes a computational invariance characteristic of the transformer architecture, which allows for the application of an orthogonal transformation to the output of one component, provided that this transformation is reversed in the subsequent component. A fundamental insight of this method is that the RMSNorm operation, conducted between network blocks, does not interfere with the transformation because the operations are commutative. Initially, the method elucidates how this invariance manifests within transformer networks connected by RMSNorm. Subsequently, it employs Principal Component Analysis (PCA) to calculate transformations at each layer, ensuring that the signal between blocks is projected onto its principal components. Ultimately, it eliminates the less significant principal components, effectively trimming rows or columns from the altered network.

- **Wanda** unifies magnitude-based and gradient-based importance criteria into a single framework, enabling accurate identification of structurally redundant components, such as attention heads, neurons, and layers, without requiring costly retraining. By iteratively removing the least important structures and lightly fine-tuning the remaining model, the approach significantly reduces model size and inference cost while preserving downstream performance

## C  The Detailed Process of Data Collection

Since we are the first to do adaptive pruning for LLMs, we are met with the challenge of an absence of pre-existing datasets for this purpose. To ensure a comprehensive dataset, we generated 100 base pruning configurations by employing a broad range of 12 distinct heuristic methods. These heuristics can be categorized as follows:

- **Exploratory Sampling** (50 configurations): To ensure wide coverage of the search space and prevent bias from specific strategies, we also included 2 random sampling heuristics (Random-Layer and Random-Block), generating 25 configurations from each.

- **Knowledge-Driven Methods** (50 configurations): We used 10 methods that leverage specific criteria. These include our 3 custom heuristics (PPL-based, Energy-based, and Score-based) and 7 established methods which also served as baselines (L2, ShortGPT, SliceGPT, Channel, LLMPruner, Wanda, LoRA-Prune). We generated 5 different configurations from each of these 10 methods.

To further enrich the dataset for robust training, we then applied a 10x Mixup data augmentation (Zhang et al., 2018), expanding these 100 base points into the 1000 "pruning-score" pairs used to train `Alps`. We will add these specific details to the main paper to make our process fully transparent.

## D  Additional Results

### D.1  Examples of Generated Pruning Ratio:

A distinguishing feature of `Alps` is its ability to generate pruning configurations autonomously through a learned decoder with no manual intervention. After identifying the optimal representation in continuous space through gradient-steered optimization, our LSTM decoder generates layer-specific pruning ratios directly from this representation. This process is determined entirely by the model's learned parameters, requiring no human guidance or post-processing.

For Llama2-7B with overall 20% pruning ratio, the specific pruning configuration applied was *[0.00, 0.31, 0.00, 0.00, 0.17, 0.00, 0.00, 0.00, 0.00, 0.17, 0.00, 0.00, 0.2, 0.31, 0.08, 0.49, 0.29, 0.00, 0.29, 0.18, 0.41, 0.45, 0.32, 0.46, 0.32, 0.48, 0.37, 0.00, 0.00, 0.47, 0.42, 0.00]*.

For Llama2-7B with overall 50% pruning ratio, the specific pruning configuration applied was *[0.00, 0.00, 0.00, 0.00, 0.00, 0.00, 0.00, 0.00, 0.79, 0.92, 0.7, 0.91, 0.88, 0.91, 0.89, 0.72, 0.00, 0.57, 0.00, 0.88, 0.85, 0.66, 0.89, 0.82, 0.97, 0.91, 0.97, 0.68, 0.00, 0.81, 0.00, 0.00]*.

These configurations show sophisticated non-uniform patterns that adapt to the specific importance of different layers. The 20% pruning case preserves many early layers completely while applying varied pruning to later layers. The 50% case shows a more aggressive but still structured approach, preserving critical layers throughout the network. Such adaptive, layer-specific patterns would be

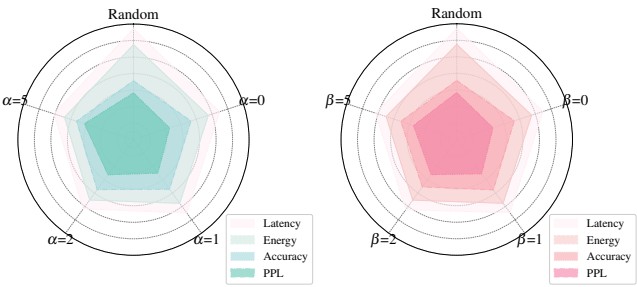

Figure 9: Hyperparameter sensitivity of Alps. Left: $\alpha$; Right: $\beta$. Evaluate the model performance, energy consumption, and latency of the pruned model with different values.

extremely difficult to discover through manual tuning or predefined heuristics, demonstrating the advantage of our gradient-steered optimization in continuous representation space.

## D.2 HYPERPARAMETRIC ANALYSIS

Alps incorporates two penalty factors, $\alpha$ and $\beta$, to balance latency and energy during pruning. As shown in Figure 9, Alps maintains robust performance across a range of $\alpha/\beta$ values. Larger values emphasize system efficiency, while smaller ones favor task accuracy, enabling adaptive trade-offs. Notably, Alps exhibits consistent behavior across all non-zero settings, with $\alpha = \beta = 2$ yielding slightly better results. These findings highlight the framework's insensitivity to hyperparameter tuning and offer guidance for practical deployment.

## D.3 MORE DOWNSTREAM TASKS.

MMLU (Hendrycks et al., 2021), covering 57 tasks across diverse domains like mathematics, history, law, and ethics. (Table 2)

Table 2: Five-shot MMLU performance of the pruned Llama2-7B (overall pruning ratio = 20%) across different pruning schemes.

| Method | STEM | Sciences | Humanities | Other | Avg |
|---|---|---|---|---|---|
| Random-Layer | 21.78 | 24.00 | 23.37 | 24.34 | 23.37 |
| PPL-Layer | 22.77 | 28.00 | 29.31 | 34.60 | 28.67 |
| Random-Block | 19.14 | 25.85 | 24.75 | 28.74 | 24.62 |
| L2 | 27.06 | 23.69 | 24.75 | 24.34 | 24.96 |
| ShortGPT | **32.67** | 23.69 | 25.94 | 24.63 | 26.73 |
| SliceGPT | 27.06 | 27.38 | 27.33 | 34.02 | 28.94 |
| Channel | 27.72 | 27.69 | 28.12 | 27.27 | 27.70 |
| LLMPruner | 22.77 | 27.69 | 26.93 | 26.39 | 25.94 |
| **Alps** | 23.1 | **29.23** | **29.70** | **35.48** | **29.38** |

## D.4 MORE BASE MODELS AND PRUNING RATIO.

**(3.1) Llama2-13B with 50% pruning ratio.** (Table 3)

**(3.2) Vicuna-7B with 50% pruning ratio.** (Table 4)

**(3.3) Llama-7B with 20% pruning ratio.** (Table 5)

**(3.4) Llama-7B with 50% pruning ratio.** (Table 6)

**(3.5) Llama3-8B with 20% and 50% pruning ratio.** (Table 7)

**(3.6) Qwen1.5-7B with 20% and 50% pruning ratio.** (Table 8)

**(3.6) Qwen1.5-7B with 20% and 50% pruning ratio.** (Table 9)

Table 3: Zero-shot performance (7 datasets) and perplexity (2 datasets) of the pruned Llama2-13B (overall pruning ratio = 50%)

| Method | Perplexity ↓ | | Downstream task performance (%) ↑ | | | | | | | |
| | WikiText2 | PTB | BoolQ | PIQA | OBQA | ARC-c | ARC-e | WinoGrande | HellaSwag | Average |
|---|---|---|---|---|---|---|---|---|---|---|
| Random-Layer | 3539.98 | 4474.95 | 41.38 | 52.45 | 28.60 | 27.22 | 26.09 | 48.70 | 25.78 | 35.75 |
| PPL-Layer | 40.02 | 159.84 | 56.57 | 63.66 | 33.40 | 27.99 | 43.52 | 53.43 | 43.72 | 46.04 |
| ShortGPT | 48.75 | 189.08 | 61.74 | 63.49 | 33.20 | 27.47 | 41.96 | 51.85 | 43.39 | 46.16 |
| Random | 363.04 | 409.78 | 60.55 | 54.08 | 28.40 | 26.37 | 28.28 | 49.80 | 34.68 | 40.31 |
| L2 | 81.96 | 295.15 | 42.20 | 62.35 | 30.20 | 28.16 | 42.59 | 50.67 | 31.46 | 41.09 |
| SliceGPT | 74.95 | 223.22 | 56.54 | 59.96 | 28.80 | 27.65 | 41.71 | 54.85 | 40.02 | 44.36 |
| Channel | 81.96 | 295.15 | 62.23 | 61.04 | 30.40 | 25.09 | 36.45 | 50.83 | 37.44 | 43.35 |
| LLMPruner | 37.09 | 126.94 | 50.43 | 68.12 | 35.00 | 29.10 | 49.45 | 52.17 | 48.13 | 47.48 |
| **Alps** | **25.53** | **108.15** | **62.23** | **70.46** | **35.00** | **30.97** | **56.02** | 53.28 | **53.24** | **51.60** |

Table 4: Zero-shot performance (7 datasets) and perplexity (2 datasets) of the pruned Vicuna-7B (overall pruning ratio = 50%)

| Method | Perplexity ↓ | | Downstream task performance (%) ↑ | | | | | | | |
| | WikiText2 | PTB | BoolQ | PIQA | OBQA | ARC-c | ARC-e | WinoGrande | HellaSwag | Average |
|---|---|---|---|---|---|---|---|---|---|---|
| Random-Layer | 3595.72 | 3887.90 | 39.72 | 54.52 | 27.60 | 26.62 | 26.09 | 50.20 | 26.43 | 35.88 |
| PPL-Layer | 272.97 | 553.58 | 57.34 | 55.06 | 33.20 | 25.60 | 32.20 | 49.72 | 30.27 | 40.48 |
| ShortGPT | 1766.16 | 3768.28 | 41.56 | 56.80 | 32.00 | 28.75 | 30.60 | 50.12 | 33.47 | 39.04 |
| Random | 560.11 | 580.15 | 61.35 | 64.74 | 28.80 | 26.88 | 43.06 | 50.51 | 38.24 | 44.80 |
| SliceGPT | 272.68 | 1256.28 | 60.24 | 56.96 | 30.60 | 25.94 | 40.82 | 52.25 | 31.85 | 42.67 |
| Channel | 131.49 | 627.29 | 58.41 | 58.27 | 28.80 | 25.09 | 35.98 | 50.91 | 33.73 | 41.60 |
| LLMPruner | 45.98 | 146.68 | 61.90 | 66.65 | 32.60 | 26.71 | 46.72 | 50.99 | 41.60 | 46.74 |
| **Alps** | **34.17** | **100.81** | 49.27 | **67.03** | **34.00** | **29.10** | **50.34** | **54.85** | **45.42** | **47.14** |

Table 5: Zero-shot performance (7 datasets) and perplexity (2 datasets) of the pruned Llama-7B (overall pruning ratio = 20%)

| Method | Perplexity ↓ | | Downstream task performance (%) ↑ | | | | | | | |
| | WikiText2 | PTB | BoolQ | PIQA | OBQA | ARC-c | ARC-e | WinoGrande | HellaSwag | Average |
|---|---|---|---|---|---|---|---|---|---|---|
| Llama-7B | 12.62 | 53.85 | 76.09 | 78.84 | 42.60 | 42.83 | 68.94 | 67.56 | 73.95 | 64.40 |
| Random-Layer | 101.20 | 971.56 | 58.90 | 60.17 | 28.80 | 27.05 | 43.18 | 52.96 | 34.28 | 43.62 |
| PPL-Layer | 21.89 | 71.77 | 66.51 | 71.44 | 36.8 | 34.39 | 54.59 | 63.22 | 63.21 | 55.74 |
| ShortGPT | 26.45 | 89.67 | 41.50 | 70.95 | 38.00 | 33.96 | 53.11 | 61.96 | 60.28 | 51.39 |
| Random | 34.37 | 139.42 | 57.40 | 73.23 | 35.20 | 37.37 | 63.17 | 58.33 | 52.65 | 53.91 |
| L2 | 176.24 | 302.15 | 50.76 | 64.42 | 30.80 | 29.1 | 33.21 | 55.17 | 31.85 | 42.19 |
| SliceGPT | 37.59 | 208.72 | 68.72 | 69.70 | 36.60 | 37.03 | 60.19 | 62.27 | 53.38 | 55.41 |
| Channel | 20.28 | 80.06 | 63.21 | 73.94 | 31.80 | 38.05 | 62.16 | 62.12 | 61.24 | 56.07 |
| LLMPruner | 17.45 | 77.90 | 54.62 | 77.48 | 40.00 | 38.91 | 69.07 | 61.72 | 68.76 | 58.65 |
| Wanda | 18.43 | 33.16 | 65.75 | 74.70 | 39.40 | 36.26 | 60.65 | 59.35 | 64.52 | 57.23 |
| LoraPrune | 16.8 | 28.75 | 65.62 | 79.31 | 39.14 | 37.69 | 65.87 | 62.76 | 70.00 | 60.05 |
| **Alps** | **16.10** | **68.21** | **71.59** | 75.57 | **40.60** | **40.36** | 63.55 | **66.61** | **69.76** | **61.15** |

Table 6: Zero-shot performance (7 datasets) and perplexity (2 datasets) of the pruned Llama-7B (overall pruning ratio = 50%)

| Method | Perplexity ↓ | | Downstream task performance (%) ↑ | | | | | | | |
| | WikiText2 | PTB | BoolQ | PIQA | OBQA | ARC-c | ARC-e | WinoGrande | HellaSwag | Average |
|---|---|---|---|---|---|---|---|---|---|---|
| Random-Layer | 14445.31 | 23816.29 | 49.57 | 53.21 | 29.6 | 28.07 | 26.22 | 49.25 | 25.84 | 37.39 |
| PPL-Layer | 132.52 | 302.15 | 61.07 | 56.04 | 30.60 | 24.66 | 30.93 | 50.20 | 33.04 | 40.93 |
| ShortGPT | 808.61 | 1322.79 | 42.02 | 55.39 | 29.40 | 28.50 | 29.12 | 51.46 | 33.21 | 38.44 |
| Random | 10819.04 | 12259.57 | 40.03 | 53.37 | 27.60 | 28.07 | 26.94 | 49.41 | 26.04 | 35.92 |
| L2 | 10819.04 | 1071.23 | 59.66 | 60.5 | 28.60 | 27.05 | 30.13 | 49.96 | 27.03 | 40.42 |
| SliceGPT | 4010.22 | 9586.37 | 39.20 | 53.32 | 27.60 | 24.66 | 28.37 | 48.54 | 27.45 | 35.59 |
| Channel | 210.94 | 624.84 | 55.57 | 58.11 | 30.20 | 24.66 | 33.59 | 52.09 | 31.63 | 40.84 |
| LLMPruner | 41.05 | 158.60 | 60.21 | 64.42 | 31.80 | 27.39 | 43.60 | 52.09 | 42.05 | 45.94 |
| **Alps** | **32.86** | **111.16** | 59.79 | **66.16** | **33.40** | **28.50** | **49.07** | **54.70** | **45.84** | **48.21** |

Table 7: Zero-shot performance and perplexity of pruned Llama3-8B. **ALPS** achieves the best trade-off between sparsity and performance. **Bold** and Underlined indicate the highest and second highest scores, respectively.

| Method | Perplexity ↓ | | | Downstream task performance (%) ↑ | | | | | | | |
|---|---|---|---|---|---|---|---|---|---|---|---|
| | WikiText2 | PTB | Average | BoolQ | PIQA | OBQA | ARC-c | ARC-e | WinoG | HellaSwag | Average |
| Llama3-8B | 6.13 | 9.91 | 8.02 | 81.35 | 80.78 | 45.00 | 53.33 | 77.69 | 72.61 | 79.14 | 69.99 |
| **Llama3-8B (20%)** | | | | | | | | | | | |
| Random-Layer | 4892.45 | 3812.10 | 4352.28 | 44.12 | 53.40 | 28.10 | 29.05 | 27.15 | 51.50 | 27.30 | 37.23 |
| PPL-Layer | 35.67 | 89.20 | 62.44 | 60.15 | 71.50 | 36.80 | 35.40 | 56.20 | 60.10 | 62.50 | 54.66 |
| Random-Block | 102.34 | 245.10 | 173.72 | 60.55 | 68.45 | 32.10 | 30.50 | 52.80 | 50.15 | 40.20 | 47.82 |
| ShortGPT | 118.62 | 205.30 | 161.96 | 65.02 | 71.00 | 33.20 | 42.41 | 56.65 | 70.88 | 64.61 | 57.68 |
| SliceGPT | 45.12 | 192.40 | 118.76 | 73.40 | 71.20 | 35.80 | 39.10 | 60.15 | 62.50 | 61.20 | 57.62 |
| LLMPruner | 23.21 | 98.40 | 60.81 | 67.68 | 75.03 | 36.00 | 37.03 | 61.28 | 60.77 | 57.76 | 56.51 |
| Wanda | 29.92 | 92.15 | 61.04 | 59.42 | 77.31 | 39.60 | 39.68 | 67.68 | 59.67 | 58.77 | 57.45 |
| **Alps** | **19.85** | **85.40** | **52.63** | **74.15** | **77.95** | **40.10** | **43.20** | **68.50** | **71.20** | **65.10** | **62.89** |
| **Llama3-8B (50%)** | | | | | | | | | | | |
| Random-Layer | 11502.10 | 9820.50 | 10661.30 | 38.50 | 53.20 | 28.50 | 28.10 | 26.30 | 50.50 | 26.10 | 35.89 |
| PPL-Layer | 210.45 | 620.10 | 415.28 | 49.10 | 56.80 | 31.50 | 27.80 | 32.10 | 51.60 | 31.50 | 40.06 |
| Random-Block | 580.40 | 498.17 | 630.30 | 60.15 | 61.80 | 29.50 | 27.10 | 41.20 | 50.60 | 38.50 | 44.12 |
| ShortGPT | 4135.73 | 2150.20 | 3142.97 | 60.86 | 55.39 | 29.60 | 28.84 | 29.71 | 54.54 | 29.38 | 41.19 |
| SliceGPT | 320.15 | 1102.50 | 711.33 | 63.45 | 59.20 | 28.10 | 26.90 | 42.10 | 52.30 | 37.10 | 44.16 |
| LLMPruner | 125.91 | 310.20 | 218.06 | 52.63 | 56.37 | 28.60 | 22.53 | 36.07 | 50.43 | 31.08 | 39.67 |
| Wanda | 133.29 | 205.40 | 169.35 | 58.75 | 60.88 | 26.20 | 23.04 | 37.84 | 50.67 | 31.87 | 41.32 |
| **Alps** | **115.50** | **190.20** | **152.85** | **64.10** | **63.50** | **30.80** | **30.15** | **43.50** | **55.10** | **39.40** | **46.65** |

Table 8: Zero-shot performance and perplexity of pruned Qwen1.5-7B. **ALPS** achieves the best trade-off between sparsity and performance. **Bold** and Underlined indicate the highest and second highest scores, respectively.

| Method | Perplexity ↓ | | | Downstream task performance (%) ↑ | | | | | | | |
|---|---|---|---|---|---|---|---|---|---|---|---|
| | WikiText2 | PTB | Average | BoolQ | PIQA | OBQA | ARC-c | ARC-e | WinoG | HellaSwag | Average |
| Qwen1.5-7B | 7.95 | 11.93 | 9.94 | 82.45 | 79.05 | 41.60 | 42.83 | 62.25 | 66.14 | 76.90 | 64.46 |
| **Qwen1.5-7B (20%)** | | | | | | | | | | | |
| Random-Layer | 6102.50 | 5210.40 | 5656.45 | 40.15 | 51.20 | 24.50 | 22.10 | 25.40 | 49.80 | 25.60 | 34.11 |
| PPL-Layer | 42.10 | 98.50 | 70.30 | 61.20 | 68.50 | 32.40 | 30.15 | 48.60 | 55.40 | 58.20 | 50.64 |
| Random-Block | 145.20 | 280.60 | 212.90 | 59.80 | 65.20 | 30.10 | 28.50 | 45.20 | 51.20 | 42.50 | 46.07 |
| ShortGPT | 130.50 | 220.10 | 175.30 | 64.50 | 69.80 | 31.80 | 35.60 | 52.10 | **64.50** | 60.80 | 54.16 |
| SliceGPT | 55.40 | 210.50 | 132.95 | **71.20** | 69.50 | 32.50 | 33.40 | 53.80 | 58.20 | 58.90 | 53.93 |
| LLMPruner | 28.50 | 115.60 | 72.05 | 66.10 | 71.40 | 35.80 | 31.50 | 55.60 | 59.40 | 61.50 | 54.47 |
| Wanda | 35.60 | 105.40 | 70.50 | 58.40 | 73.20 | 35.10 | 32.80 | 58.90 | 61.20 | 57.20 | 53.83 |
| **Alps** | **22.40** | **95.30** | **58.85** | 70.80 | **74.50** | **36.50** | **36.20** | **60.10** | 63.80 | **62.50** | **57.77** |
| **Qwen1.5-7B (50%)** | | | | | | | | | | | |
| Random-Layer | 14205.10 | 12050.60 | 13127.85 | 35.40 | 50.10 | 23.50 | 21.20 | 24.50 | 48.20 | 24.10 | 32.43 |
| PPL-Layer | 255.40 | 710.20 | 482.80 | 46.50 | 54.20 | 28.50 | 24.60 | 28.90 | 50.10 | 29.50 | 37.47 |
| Random-Block | 720.50 | 610.80 | 665.65 | 58.40 | 59.50 | 27.20 | 23.50 | 38.60 | 49.50 | 35.40 | 41.73 |
| ShortGPT | 5210.40 | 2850.10 | 4030.25 | 58.90 | 53.40 | 28.40 | **25.80** | 28.50 | 52.80 | 28.10 | 39.41 |
| SliceGPT | 410.20 | 1250.60 | 830.40 | **61.50** | 56.80 | 28.50 | 24.20 | **40.50** | 50.20 | 35.10 | 42.40 |
| LLMPruner | 145.20 | 350.50 | 247.85 | 50.20 | 54.80 | 25.80 | 21.50 | 32.40 | 49.60 | 36.80 | 38.73 |
| Wanda | 158.60 | 240.80 | 199.70 | 54.50 | 58.90 | 25.20 | 22.10 | 35.20 | 50.80 | 30.50 | 39.46 |
| **Alps** | **135.50** | **225.40** | **180.45** | 60.20 | **60.50** | **29.10** | 23.60 | 39.80 | **53.50** | **38.20** | **42.99** |

# E  ADDITIONAL DISCUSSION

## E.1  FOR STATISTICAL SIGNIFICANCE

To balance experimental scale with computational feasibility, we adopted a standard protocol from recent large-scale LLM pruning research Ma et al. (2023); Zhang et al. (2023). Specifically, we utilized a fixed random seed (42) for all experiments, applying it consistently across our method and all baselines to ensure a direct and fair comparison. The consistent and significant performance advantage of Alps across a wide array of models, tasks, and settings provides strong evidence for the robustness of our conclusions.

To ensure reproducibility and fair comparison, we followed the standard practice established by prior work in this area. We used a fixed random seed (42) for all our experiments. Crucially, this same seed was applied to all baseline methods to ensure a direct and fair comparison. This approach, focusing on reproducibility with a fixed seed, is consistent with the evaluation protocol of works we compare against.

Table 9: Zero-shot performance and perplexity of pruned Qwen2.5-7B. **ALPS** achieves the best trade-off between sparsity and performance. **Bold** and Underlined indicate the highest and second highest scores, respectively.

| Method | Perplexity ↓ | | | Downstream task performance (%) ↑ | | | | | | | |
|---|---|---|---|---|---|---|---|---|---|---|---|
| | WikiText2 | PTB | Average | BoolQ | PIQA | OBQA | ARC-c | ARC-e | WinoG | HellaSwag | Average |
| Qwen2.5-7B | 6.85 | 11.36 | 9.11 | 84.61 | 79.71 | 47.40 | 51.01 | 77.40 | 73.00 | 78.95 | 70.30 |
| Qwen2.5-7B (20%) | | | | | | | | | | | |
| Random-Layer | 5820.10 | 4910.30 | 5365.20 | 42.15 | 54.30 | 26.20 | 24.10 | 27.50 | 50.40 | 28.10 | 36.11 |
| PPL-Layer | 38.40 | 85.20 | 61.80 | 65.40 | 70.20 | 36.50 | 34.20 | 52.60 | 58.40 | 62.10 | 54.20 |
| Random-Block | 120.50 | 255.40 | 187.95 | 62.10 | 68.50 | 34.10 | 31.50 | 49.80 | 54.20 | 45.60 | 49.40 |
| ShortGPT | 115.20 | 198.50 | 156.85 | 68.50 | 71.40 | 35.80 | 38.20 | 56.40 | **69.50** | 64.20 | 57.71 |
| SliceGPT | 48.60 | 185.20 | 116.90 | **76.40** | 72.10 | 37.40 | 37.50 | 58.20 | 62.10 | 63.50 | 58.17 |
| LLMPruner | 24.10 | 92.40 | 58.25 | 70.20 | 74.50 | 40.50 | 35.60 | 60.40 | 64.20 | 66.80 | 58.89 |
| Wanda | 31.50 | 88.60 | 60.05 | 62.50 | 76.80 | 39.20 | 38.50 | 62.10 | 65.40 | 64.50 | 59.86 |
| **Alps** | **18.20** | **75.40** | **46.80** | 75.20 | **77.50** | **41.20** | **42.10** | **66.50** | 64.80 | **68.40** | **62.24** |
| Qwen2.5-7B (50%) | | | | | | | | | | | |
| Random-Layer | 13502.50 | 11204.20 | 12353.35 | 37.20 | 51.50 | 24.80 | 22.50 | 25.10 | 49.20 | 25.40 | 33.67 |
| PPL-Layer | 210.50 | 650.40 | 430.45 | 48.50 | 55.40 | 29.20 | 25.40 | 30.20 | 51.50 | 30.50 | 38.67 |
| Random-Block | 650.40 | 580.20 | 615.30 | 60.20 | 61.50 | 28.50 | 24.50 | 40.20 | 51.20 | 37.40 | 43.36 |
| ShortGPT | 4850.20 | 2650.40 | 3750.30 | 60.50 | 56.20 | 29.50 | **28.40** | 32.50 | 55.40 | 30.20 | 41.81 |
| SliceGPT | 380.50 | 1150.20 | 765.35 | 65.40 | 56.20 | 29.10 | 26.50 | **45.20** | 52.60 | 38.50 | 45.11 |
| LLMPruner | **135.20** | 310.50 | 222.85 | 53.40 | 57.20 | 30.50 | 23.60 | 35.40 | 51.50 | 39.20 | 41.54 |
| Wanda | 142.60 | **215.40** | **179.00** | 56.20 | 60.40 | 28.40 | 24.10 | 38.50 | 53.20 | 34.50 | 42.19 |
| **Alps** | 165.40 | 225.80 | 195.60 | 62.50 | **63.80** | **32.40** | 27.80 | 43.50 | **56.10** | **41.50** | **46.80** |

## E.2 FOR OVERHEAD ANALYSIS

The overhead of constructing 100 pruning-score pairs is a modest, one-time cost. It is negligible compared to the long-term online inference deployment (Feurer et al., 2022; He et al., 2018; Jacob et al., 2017; Han et al., 2016). This data collection is performed only once during pretraining and does not affect deployment. After training, Alps rapidly generates diverse, high-quality pruning configurations in under 1 second, enabling efficient and scalable inference without further data collection or retraining. This design ensures a minimal upfront investment for a highly efficient and scalable tool.

## E.3 FOR RECOVERY

In the recovery stage, we adhere to the methodology outlined in (Hu et al., 2021). During our experiments, we configure the rank $d$ to be 8. We set the learning rate at $1 \times 10^{-5}$ with 100 warming steps. Our training sessions utilize a batch size of 64, employing the AdamW optimizer. We fine-tune for 2 epochs. This is a common practice for post-pruning recovery, consistent with prior works (Ma et al., 2023; Zhang et al., 2023). This brief, offline recovery stage takes approximately 2-3 hours on a single NVIDIA A40 GPU.

## E.4 ROBUSTNESS OF BEAM SEARCH DECODING

To mitigate the nvalid or suboptimal pruning configurations of beam search decoding. The decoder's output vocabulary consists of valid pruning ratios within [0, 1], and the generation process produces a sequence of exactly k ratios corresponding to the k layers. Therefore, out-of-range or incorrectly formatted configurations are not possible. We have a robust, multi-step strategy to ensure high-quality outputs, which is a core feature of Alps 1) **Guided Search**: The model is trained on high-performing "pruning-score" pairs, so beam search is naturally biased toward generating effective configurations. 2) **Multi-Start Optimization**: Crucially, we do not rely on a single generation. As detailed in Section 3.4, we start from the top-K known configurations (K=25 in our experiments). This optimization is repeated for all K starting points to find multiple candidate representations. 3) **Final Selection**: We then decode these candidates and use our trained evaluator $e$ to select the one with the highest estimated score for final deployment. This comprehensive search-and-select process, rather than a single guess, is our key mitigation strategy against suboptimal results.

### E.5 FOR LEARNED PRUNING SPACE

The conceptual rigor is supported by two pillars: 1) **Pragmatic Design for a Complex Space:** We did not assume a simple, convex landscape. Instead, our design inherently handles potential complexity by employing a multi-start optimization strategy, launching K parallel searches from the Top-K best-known configurations. This robustly explores high-potential regions of the space. Our optimization itself uses standard methods (Adam optimizer, MSE+NLL loss) without special tricks, ensuring the results are due to the framework's merit. 2) **Consistent Empirical Success as Evidence**: The strongest evidence for a well-behaved learned space is the consistent, successful convergence across a wide range of LLMs, datasets, and pruning ratios. It is worth emphasizing that all our reported experiments converged normally; not a single experimental group encountered convergence difficulties or was filtered out due to non-convergence. If the space were chaotic, such a standard optimization approach would not have yielded universally positive results. This macro-level success empirically validates that our framework learns a smooth and optimizable representation.

### E.6 FOR SINGLE-LAYER LSTMS ARCHITECTURE

Regarding the architecture choice, we selected a single-layer LSTM for its proven simplicity and efficiency in modeling sequential data. For our task of modeling pruning ratio sequences (which are short, with length equal to the number of LLM layers), a more complex Transformer architecture did not yield significant performance gains but incurred higher training costs and data requirements. Importantly, our framework is model-agnostic, and the encoder/decoder can be easily swapped with other sequence models. Our primary contribution lies in the encoder-evaluator-decoder framework itself, rather than a specific instantiation of it. A simple LSTM can struggle with information bottlenecks and long-term dependencies, even in short sequences (Luong et al., 2015a; Bahdanau et al., 2016). Attention allows the decoder to dynamically focus on the most relevant encoded layer information when generating each new pruning ratio. We found that removing the attention mechanism caused the sequence reconstruction loss (seq2seq NLL) to increase by 30-50%. This confirms that attention plays a crucial role in accurately capturing inter-layer dependencies and is a necessary component for our decoder's performance.

## F LIMITATION

While `Alps` demonstrates strong adaptability and efficiency, several limitations warrant discussion. First, its performance is contingent on the quality and diversity of the heuristic-based data used for training. The framework's objective is therefore not to guarantee a global optimum, but to learn a smooth and effective search space defined by these initial methods. Crucially, our experiments validate this approach by showing that optimizing within this learned space consistently discovers configurations that outperform any single heuristic in the training set. A related challenge is the computational cost of generating this initial dataset. Second, Alps is trained per architecture and does not yet generalize across model families (e.g., from LLaMA-2 to LLaMA-3). Lastly, our current formulation targets structured pruning; extending `Alps` to hybrid or unstructured settings could further broaden its applicability.

