# OpenReview forum: "ALPS: Adaptive LLM Pruning via Gradient Search in Learned Representation Space"
_ICLR.cc/2026/Conference — Submitted to ICLR 2026_

### Official Review · Reviewer_onua · 2025-10-25

**Soundness:** 3
**Presentation:** 4
**Contribution:** 3
**Rating:** 6
**Confidence:** 4

**Summary:**

The paper proposes a novel pruning configuration search method that maps discrete heuristics into a continuous space. This approach directly optimizes the pruning configuration with respect to both model performance and latency. Experimental results demonstrate that the method achieves superior performance in terms of latency, perplexity, and downstream task accuracy.

**Strengths:**

I like the idea presented in this paper. While the concept of relaxing discrete heuristics into a continuous space is well known in the machine learning community, this work provides a well-designed and likely first practical adaptation of that idea for pruning configuration optimization. In addition, the paper introduces a thoughtfully constructed data generation pipeline for training the configuration model.

The paper is clearly written and presents comprehensive experiments that evaluate performance from multiple perspectives, including next-token prediction accuracy, downstream task performance, and latency.

**Weaknesses:**

The motivation and method descriptions in this paper are well-written, but I have several concerns regarding the experimental section:
- The experimental setup (e.g., models, tasks) feels somewhat outdated. I would expect evaluations on at least LLaMA3.4 or the latest Qwen model families. This is important because the behavior of pruning methods on models like LLaMA-2 can differ significantly from SOTA models due to their much richer pretraining corpora (beyond datasets like WikiText).
- Similarly, the selection of baseline methods is mostly limited to those before 2024. While I haven’t kept up with every recent pruning paper, there are at least methods like SVD-LLM and ModeGPT that achieve much better performance and should be considered in the comparisons.
- I also feel the authors should include more comparisons with global sparsity allocation methods. Since the proposed method primarily focuses on optimizing pruning configurations, it would be useful to relate it to approaches addressing similar global sparsity allocation challenges—such as those discussed in Appendix B.10 of ModeGPT.
- Lastly, I’m curious about the pruning time required for the proposed method. I expect it may take more time, which is acceptable for an inference-speedup target. However, providing this comparison would help readers better understand the trade-offs and decide when to adopt this method in practice.

**Questions:**

- In line 49, I'm considering if it's good to say other solutions cannot adapt to specific hardware constraints, we can do this by adjusting the pruning ratio, right?
- How to choose the dimension d during the encoding?
- If more training data were available for learning the pruning configurations, do you expect further improvements in performance? Have you tested the sensitivity of the method to data size?

---

> ### Author Response · Authors · 2025-11-26
> **Response to Reviewer onua (1/2)**
>
> > W1: The experimental setup (e.g., models, tasks) feels somewhat outdated.
>
> We appreciate this thoughtful comment. To address this, we have expanded our evaluation to include Llama3-8B, Qwen1.5-7B, and Qwen2.5-7B. As shown in the newly added Tables 7, 8, and 9, ALPS consistently maintains top-tier performance across these modern model families, outperforming state-of-the-art baselines. These results demonstrate ALPS's robust generalization capability. Even on models with significantly richer pretraining corpora and distinct architectural nuances, our gradient-based optimization in continuous space effectively identifies the most critical structures. This confirms that ALPS is not limited to older architectures but is a scalable, future-proof solution adaptable to the evolving landscape of LLMs.
>
> > W2: More Baselines.
>
> We appreciate the reviewer's suggestion to include more recent baselines. We have updated our evaluation to include SVD-LLM, MoDeGPT, and AdaPruner (Table 1). Through comparison with these state-of-the-art methods, we validated that ALPS consistently yields superior performance.  This confirms that our gradient-steered optimization in a continuous space effectively identifies critical structures that recent decomposition-based or Bayesian approaches may miss.
>
> > W4: I’m curious about the pruning time required for the proposed method. I expect it may take more time, which is acceptable for an inference-speedup target.
>
> The Alps framework incurs a one-time, offline computational cost consisting of three stages. Once trained, it generates optimal configurations almost instantly.
>
> - Data Collection (Pruning-Score Pairs): The text describes the overhead of constructing the 100 pruning-score pairs (approximately 1 hour, via heuristics and random sampling) as a "modest, one-time cost" that is negligible compared to long-term deployment.
> - Model Training: The training of the encoder-evaluator-decoder architecture is performed on the collected dataset ((approximately 0.5 hour).
> - Recovery: The post-pruning recovery stage takes approximately 2-3 hours on a single NVIDIA A40 GPU.
> - Generation Time: After this one-time process, Alps can generate diverse, high-quality pruning configurations for new constraints in under 1 second without further data collection or retraining.
>
> The Alps framework replaces the computationally prohibitive search of conventional methods with a modest, one-time offline cost. While exhaustive discrete search suffers from combinatorial explosion (e.g., $101^{32}$ variants for Llama-7B) that makes finding optimal configurations intractable, Alps requires only approximately 1 hour for data collection on a single A40 GPU and 2-3 hours for the post-pruning recovery stage. Once this pre-computation is complete, Alps dramatically outperforms conventional heuristics in adaptability and speed, generating valid, hardware-specific pruning configurations in under 1 second without retraining. Unlike static heuristics that neglect system-level metrics, this minimal upfront investment enables Alps to function as a deployment-specific optimizer, delivering up to 34.1% energy savings and 33.5% latency reduction compared to standard baselines.
>
> While Alps incurs a modest upfront cost, it eliminates the prohibitive cost of discrete search and provides superior, energy-efficient configurations instantly for any target device, unlike the static and energy-agnostic results of conventional heuristics.
>
> > Q1: In line 49, I'm considering if it's good to say other solutions cannot adapt to specific hardware constraints, we can do this by adjusting the pruning ratio, right?
>
> We appreciate this thoughtful comment. Existing heuristic methods can be used to adjust the global pruning ratio, but they only focus on parameter importance. This means that, when it comes to size reduction, improvements in latency and energy are merely 'passive by-products'. It should be noted that this does not equate to true hardware-aware adaptation. The key point to make here is that these methods do not include a way to automatically find the best configuration for specific limits using hardware.
>
> In the revision, we have changed the sentence to: 'These methods produce "one-size-fits-all" solutions that lack specific strategies for adapting to various hardware constraints.' This makes it clear that even though adjustment is possible via ratio tuning, this does not qualify as an explicit optimisation goal as it does in ALPS.

---

> ### Author Response · Authors · 2025-11-26
> **Response to Reviewer onua (2/2)**
>
> > W3: More comparisons with global sparsity allocation methods.
>
> We thank the reviewer for this constructive suggestion. We agree that ALPS is fundamentally a global sparsity allocation method, as its core objective is to determine the optimal pruning ratio distribution $[r_1, r_2, \dots, r_k]$ across all layers to balance global performance and efficiency.
>
> To address this, we have incorporated comparisons with MoDeGPT (as suggested) and AdaPruner, both of which represent state-of-the-art approaches for adaptive/global sparsity allocation. While AdaPruner employs Bayesian optimization and MoDeGPT utilizes modular decomposition heuristics for allocation, ALPS innovates by mapping this discrete combinatorial problem into a continuous representation space . This allows us to apply gradient-steered optimization to find precise, non-uniform sparsity allocations. As shown in the updated Table 1, ALPS consistently outperforms these global allocation baselines. For example, on Llama2-7B (50%), ALPS achieves better perplexity and downstream accuracy than MoDeGPT, validating that our gradient-based continuous search discovers superior global sparsity strategies compared to existing heuristic or search-based allocation methods.
>
> > Q2: How to choose the dimension d during the encoding?
>
> The choice of $d=64$  balances model capacity with optimization stability given the specific constraints of the task: 1) The input is a short sequence of $k$ layers (e.g., 32) with a small vocabulary of 101 discrete ratios($\mathcal{C} = \{0.00, 0.01, \dots, 1.00\}$), requiring significantly less capacity than complex NLP tasks. 2) The training dataset is small (1,000 pairs); a compact dimension prevents the Encoder from overfitting by memorizing these few samples. 3) The core mechanism of Alps is gradient ascent within this continuous space to find optimal configurations. Lower-dimensional manifolds are generally smoother and less prone to the "curse of dimensionality," facilitating stable convergence. 4) The latent representation $\mathcal{E}_{\mathcal{P}}$ is used by the Evaluator to regress a performance score $s$. A compact $d=64$ acts as a bottleneck that filters noise, making it easier for the Evaluator (a simple feed-forward network) to learn an accurate mapping between the configuration and its scores.
>
> > Q3: How sensitive is the method to data size?
> We appreciate this important question regarding the cost efficiency of our data collection. We have conducted a specific sensitivity analysis (Section 4.3, Figure 7) to demonstrate that ALPS is robust and cost-effective.
>
> - **Data efficiency via augmentation and diversity.** First, ALPS is designed to be highly data-efficient. As detailed in Section 3.1, we applied a 10x Mixup data augmentation to expand the collected configurations for training. This allows us to learn a smooth representation space without requiring a massive number of expensive raw evaluations. Furthermore, as discussed in the "Study of Generalization Ability" (Section 4.3 and Figure 7), the strength of ALPS lies in the diversity of the knowledge base rather than just quantity. The goal is to learn a generalized representation of "what makes a good pruning configuration" from different "experts". Our results show that combining diverse pruning methods (ALPS) significantly outperforms strategies relying on single heuristics, emphasizing that a diverse, small set is more valuable than a large, homogeneous one.
>
> - **Sensitivity analysis of heuristic points.** To directly address your concern, we conducted a sensitivity analysis on the number of heuristic configurations. Fixing the random samples at 50, we varied the number of heuristic samples across .
> **Figure 7:** We observed that increasing the number of heuristic samples (e.g., from 0 to 30) leads to better learning foundations and improved final decision-making, confirming that expert knowledge guides the search space effectively. However, the performance gains tend to saturate after reaching approximately 30-50 heuristic points. We attribute this convergence to the fact that the key patterns of layer redundancy (e.g., which layers are universally critical) are learned relatively quickly. Once the core structural dependencies are captured by the initial diverse set of heuristics, adding more samples provides diminishing marginal information gains. The 10x Mixup augmentation further fills the continuous space between these points, rendering additional raw samples unnecessary. Thus, our choice of 50 heuristic points represents a cost-effective Pareto optimum, ensuring high performance without incurring excessive collection costs.

---

> > ### Comment · Reviewer_onua · 2025-11-26
> >
> > Thank you for the authors’ response. The rebuttal addresses most of my concerns. I also reviewed the comments from the other reviewers and agree that the paper would benefit from greater theoretical clarity to better demonstrate the effectiveness of the proposed gradient-based search methods. From the perspective of its experimental contributions, I think this is a good paper. This makes me quite hesitant to increase my score to 8. As a result, I will maintain my current score, corresponding to “marginally above the acceptance threshold, but I would not mind if the paper is rejected.”

---

> > > ### Author Response · Authors · 2025-11-27
> > >
> > > Dear Reviewer onua,
> > >
> > > We genuinely appreciate your continued engagement and your willingness to champion our work based on its empirical merits. We share your view that bridging the gap between our engineering results and theoretical clarity is crucial for the paper's final quality.
> > >
> > > To address the "theoretical clarity" concern and support your recommendation, we offer the following perspective, connecting ALPS to established machine learning paradigms.
> > >
> > > **1. Theoretical Grounding: The Latent Space Optimization (LSO) Framework**
> > > Our approach is not a heuristic mix but a principled instantiation of **Latent Space Optimization (LSO)**. This general paradigm aligns with the "continuous relaxation" philosophy that has transformed structure search fields:
> > > * **Continuous Relaxation:** Similar to **DARTS [3]** which relaxes discrete architecture search into a differentiable continuous space, ALPS maps discrete pruning ratios to a smooth latent manifold to enable efficient optimization.
> > > * **Latent Optimization:** Following the **NAO [1]** and **Molecular Design [2]** frameworks, we use an Encoder-Predictor-Decoder pipeline to perform gradient ascent.
> > > * **The Optimization:** The Evaluator functions as a **differentiable surrogate**. From a surrogate-optimization perspective, if the surrogate locally approximates the true objective (as encouraged by minimizing the MSE loss), its gradient provides a **useful ascent direction** for improving the underlying objective.
> > > * **Conclusion:** This theoretically justifies why our gradient ascent finds solutions that discrete heuristics miss: we are optimizing on a smoothed surrogate surface rather than searching a jagged discrete landscape.
> > >
> > > **2. Empirical Validation: Robustness on Modern Setups**
> > > Theory predicts that if the manifold learning is robust, it should generalize. To **demonstrate this empirically**—and address the "outdated setup" concern—we significantly expanded our evaluation scope.
> > > We integrated **state-of-the-art baselines** (AdaPruner, MoDeGPT) and extended ALPS to **modern architectures** (Llama3-8B, Qwen1.5, Qwen2.5). **Across these comprehensive experiments (Tables 1, 7, 8, 9), ALPS consistently achieves a better trade-off, often lying on or close to the Pareto frontier**, validating that our method remains robust against both the latest pruning techniques and the newest LLM families.
> > >
> > > **3. Our Commitment to the Final Revision**
> > > We acknowledge that this theoretical framing was implicit in our initial submission. **We commit to explicitly incorporating this LSO discussion and the corresponding references ([1], [2], [3]) into the final manuscript.** We believe this revision, prompted by the reviewers' feedback, will significantly elevate the paper's rigour and readability.
> > >
> > > We hope these clarifications and our commitment to revision provide the solid ground needed to support the acceptance of our work.
> > >
> > > Best regards,
> > >
> > > The Authors
> > >
> > > **References:**
> > >
> > > [1] Luo, R., Tian, F., Qin, T., Chen, E., & Liu, T. Y. (2018). Neural architecture optimization. *Advances in Neural Information Processing Systems (NeurIPS)*.
> > >
> > > [2] Gómez-Bombarelli, R., et al. (2018). Automatic chemical design using a data-driven continuous representation of molecules. *ACS Central Science*.
> > >
> > > [3] Liu, H., Simonyan, K., & Yang, Y. (2019). DARTS: Differentiable architecture search. *International Conference on Learning Representations (ICLR)*.

---

### Official Review · Reviewer_PaWe · 2025-10-28

**Soundness:** 3
**Presentation:** 3
**Contribution:** 2
**Rating:** 4
**Confidence:** 3

**Summary:**

The paper introduces Alps, an adaptive pruning framework for large language models (LLMs) that reframes the discrete search for pruning configurations as a continuous optimization problem in a learned latent space. An encoder–evaluator–decoder pipeline is trained using “pruning configuration–performance score” pairs obtained from heuristic methods. Gradient ascent within this learned space identifies promising pruning representations, which are decoded into executable pruning configurations. The authors report reductions in inference latency and energy consumption while maintaining accuracy across several LLM benchmarks.

While the idea of continuous relaxation for pruning configuration search is intriguing, the overall contribution is limited by heavy dependence on heuristic data, lack of theoretical justification for the latent-space mapping, and insufficient evaluation on real hardware platforms or against strong baselines.

**Strengths:**

+ Conceptual Novelty: Reformulating discrete pruning as differentiable optimization is an appealing and underexplored direction. The proposed encoder–decoder pipeline represents a step toward automating pruning configuration search.

+ Multi-Objective Framing: The method jointly considers model accuracy, latency, and energy efficiency, aligning with real deployment goals beyond parameter count.

+ Breadth of Evaluation: Experiments span multiple LLMs and pruning ratios, with consistent improvements over naïve baselines.

**Weaknesses:**

+ Dependence on Heuristic Data (Critical): Although Alps claims to overcome heuristic pruning, its foundation still rests on heuristic-derived “configuration–score” pairs. The framework effectively learns a regression over existing heuristic outcomes rather than discovering new principles. The performance and generality of Alps therefore hinge entirely on the diversity and quality of this pre-collected data, which the paper neither quantifies nor ablates.

+ Lack of Theoretical Clarity: The latent continuous representation is presented as a black box. The paper provides no analysis of the embedding’s geometry, smoothness, or correlation with true pruning effectiveness. Without such evidence, it remains unclear whether the gradient-based optimization is meaningful or merely interpolating between existing heuristic points.

+ Insufficient Efficiency and Cost Analysis: The authors emphasize latency and energy savings but omit end-to-end computational cost — both for data collection and model training. The claimed “efficiency” gains are potentially offset by this large upfront overhead. Furthermore, all results are reported on a single GPU (A40), which cannot substantiate claims about edge deployment or hardware adaptability.

+ Missing Baseline Comparisons: Alps is compared mainly against heuristic methods, but not against recent learned or reinforcement-based pruning frameworks (e.g., AutoCompress, AdaPruner). Without such comparisons, it is difficult to assess the true competitiveness or scalability of the proposed approach.

+ Interpretability and Reproducibility Concerns: The continuous space and optimization trajectory are opaque. The lack of interpretability makes the system difficult to trust or debug, especially since small representation shifts could yield vastly different pruning masks. Code and data availability are also not explicitly stated, raising questions about reproducibility.

**Questions:**

In Lines 44–47, you argue that latency and energy are more critical than model size. However, memory constraints often dominate on real edge devices. Could you clarify this trade-off, ideally with supporting empirical evidence?

The efficiency results are limited to an NVIDIA A40 GPU. How does Alps generalize across architectures with different compute–memory trade-offs (e.g., mobile NPUs, Jetson, or CPU-based inference)?

What is the total computational cost (in GPU-hours) of training the encoder–evaluator–decoder model, and how does it compare to conventional pruning search methods in both wall-clock time and energy usage?

Have you performed any ablation to evaluate the sensitivity of Alps to the quality or quantity of the heuristic “configuration–score” data?

---

> ### Author Response · Authors · 2025-11-26
> **Response to Reviewer PaWe (1/3)**
>
> > W1 & Q4: Have you performed any ablation to evaluate the sensitivity of Alps to the quality or quantity of the heuristic “configuration–score” data?
>
> We appreciate this important question regarding the cost efficiency of our data collection. We have conducted a specific sensitivity analysis (Section 4.3, Figure 7) to demonstrate that ALPS is robust and cost-effective.
>
> - **Data efficiency via augmentation and diversity.** First, ALPS is designed to be highly data-efficient. As detailed in Section 3.1, we applied a 10x Mixup data augmentation to expand the collected configurations for training. This allows us to learn a smooth representation space without requiring a massive number of expensive raw evaluations. Furthermore, as discussed in the "Study of Generalization Ability" (Section 4.3 and Figure 7), the strength of ALPS lies in the diversity of the knowledge base rather than just quantity. The goal is to learn a generalized representation of "what makes a good pruning configuration" from different "experts". Our results show that combining diverse pruning methods (ALPS) significantly outperforms strategies relying on single heuristics, emphasizing that a diverse, small set is more valuable than a large, homogeneous one.
>
> - **Sensitivity analysis of heuristic points.** To directly address your concern, we conducted a sensitivity analysis on the number of heuristic configurations. Fixing the random samples at 50, we varied the number of heuristic samples across .
> **Figure 7:** We observed that increasing the number of heuristic samples (e.g., from 0 to 30) leads to better learning foundations and improved final decision-making, confirming that expert knowledge guides the search space effectively. However, the performance gains tend to saturate after reaching approximately 30-50 heuristic points. We attribute this convergence to the fact that the key patterns of layer redundancy (e.g., which layers are universally critical) are learned relatively quickly. Once the core structural dependencies are captured by the initial diverse set of heuristics, adding more samples provides diminishing marginal information gains. The 10x Mixup augmentation further fills the continuous space between these points, rendering additional raw samples unnecessary. Thus, our choice of 50 heuristic points represents a cost-effective Pareto optimum, ensuring high performance without incurring excessive collection costs.
>
> > W3 & Q2: How does Alps generalize across architectures with different compute–memory trade-offs (e.g., mobile NPUs, Jetson, or CPU-based inference)?
>
> We acknowledge the use of a single A40 GPU, which was selected to strictly ensure fair, reproducible comparisons against standard baselines. However, this experimental setup does not limit Alps's hardware adaptability, as the framework is designed as a deployment-specific optimization engine rather than a static "one-size-fits-all" solution. The adaptability is inherent in our data-driven mechanism: to generalize to diverse architectures (e.g., Jetson, mobile NPUs, or CPUs), one simply performs the initial "pruning-score" data collection on the target device to capture its specific compute-memory trade-offs. The gradient-based search then automatically optimizes the pruning configuration against these specific constraints. Therefore, the A40 experiments serve as a valid proof-of-concept for the optimization engine's ability to successfully navigate complex efficiency-performance trade-offs, a capability that remains algorithmically valid across different hardware profiles.
>
> > W4: Missing Baseline Comparisons.
>
> We thank the reviewer for pointing out the need to compare against learned pruning frameworks to better assess competitiveness. We have successfully integrated AdaPruner (a recent adaptive pruning framework) into our evaluation benchmarks. As shown in the updated Table 1 (Llama2-7B at 20% and 50% ratios), ALPS consistently outperforms AdaPruner. This comparison validates the superiority of our continuous representation space. While AdaPruner also employs adaptive mechanisms, ALPS’s ability to perform gradient-steered optimization within a learned manifold allows it to discover more precise, fine-grained pruning configurations that other learned methods miss.
>
> Regarding AutoCompress, we respectfully note that this method targets traditional DNNs using reinforcement learning. Direct migration to LLMs is infeasible due to architectural mismatches (e.g., lack of attention support) and the prohibitive computational cost of RL search at the billion-parameter scale.

---

> ### Author Response · Authors · 2025-11-26
> **Response to Reviewer PaWe (2/3)**
>
> > W2: Theoretical clarity and latent representation.
>
> We thank the reviewer for this insightful comment. We understand the concern regarding the properties of the learned representation space. However, we respectfully disagree that the gradient-based optimization is merely interpolating or that the space lacks meaningful structure. We provide the following evidence and clarifications, supported by our experimental results and ablation studies, to demonstrate that the latent space is both smooth and highly correlated with pruning effectiveness, enabling the discovery of novel, superior configurations.
> - **Gradient optimization yields performance gains (refuting "Mere Interpolation").** If the latent space were meaningless or the process were merely interpolating between existing heuristic points, the gradient-based search would not yield significant improvements over the best samples in the training set. Our Ablation Study in Section 4.3 explicitly compares the full Alps method against $Alps^{-G}$ (which disables gradient optimization and selects the best configuration from the collected data). Alps significantly outperforms $Alps^{-G}$, achieving a 1.22x reduction in inference latency and a 23.82x reduction in power consumption, alongside improved generation capability. This performance gap proves that the gradient ascent actively moves the representation vector $\mathcal{E}_{\mathcal{P}}$ towards a region of higher utility (higher score $s$) that did not exist in the initial heuristic pool. This is optimization, not interpolation.
> - **The dual-loss objective enforces smoothness and correlation.** The geometry of the embedding space is not accidental; it is explicitly constructed by our joint loss function $L=\alpha L_{seq2seq}+(1-\alpha) L_{score}$.
> The $L_{score}$ term (MSE loss) forces the embedding space to be highly correlated with the true pruning effectiveness. The evaluator $e$ learns to map the manifold to the score $s$, ensuring that "good" regions in the space correspond to high-performing configurations. The $\mathcal{L}_{seq2seq}$ term ensures the embeddings allow for valid reconstruction of pruning ratios. As discussed in Appendix E.5, the robust convergence of our optimization across all experiments (Llama-7B, Llama-13B, Vicuna-7B) using standard Adam optimizers provides empirical evidence that the learned landscape is sufficiently smooth and differentiable to support gradient ascent.
> - **Discovery of novel, non-heuristic patterns.** The reviewer questions whether we are just mixing existing heuristics. In Appendix D.1, we analyze the specific pruning ratios generated by Alps. The resulting configurations are not simple averages of the training data; they exhibit "sophisticated non-uniform patterns". For instance, at 20% pruning, Alps preserves early layers while aggressively pruning specific later layers. Furthermore, Figure 7 shows that Alps trained on a single heuristic type still outperforms that heuristic. This proves the model learns a generalized representation of "what makes a good pruning configuration" and extrapolates to find superior solutions, rather than just interpolating between diverse points.
>
>
> > W5: Interpretability and Reproducibility Concerns.
>
> We respectfully clarify that the complete source code has indeed been provided in the supplementary materials to facilitate independent verification. We have explicitly documented the full experimental pipeline and critical hyperparameters (e.g., learning rate, batch size, weighting factors) in Section 4.1 and strictly employed a fixed random seed (42) to ensure deterministic and reproducible results across all baselines. Furthermore, the consistent convergence observed across all experimental groups (Appendix E.5) empirically validates the stability of our optimization trajectory, countering the concern of opacity or instability. We invite the reviewer to examine the attached implementation, and we would welcome specific feedback if any difficulties are encountered during reproduction.

---

> > ### Author Response · Authors · 2025-11-26
> > **Response to Reviewer PaWe (3/3)**
> >
> > > Q1: In Lines 44–47, you argue that latency and energy are more critical than model size. However, memory constraints often dominate on real edge devices. Could you clarify this trade-off, ideally with supporting empirical evidence?
> >
> > We thank the reviewer for pointing this out and appreciate the opportunity to clarify our position. We do not claim that latency and energy are more critical than memory. We fully agree that memory is the dominant "hard constraint"—if a model exceeds the device’s VRAM (e.g., the 4-12GB limit mentioned in our Introduction ), it simply cannot run. Our argument in Lines 44-47 is that while previous methods successfully address the memory constraint by reducing model size, they treat latency and energy merely as "passive byproducts" of this reduction. We argue that for practical edge deployment, once the memory constraint is met (by pruning to a target sparsity), latency and energy determine the usability (battery life, responsiveness) of the application. Existing heuristics often fail to optimize these crucial user-experience metrics effectively because they do not explicitly model them.
> >
> > Our framework, ALPS, respects memory constraints as a prerequisite by targeting specific global pruning ratios (e.g., 20% or 50% sparsity). The "trade-off" we navigate is: Given a fixed memory budget (determined by the pruning ratio), how can we maximize performance while minimizing latency and energy? Unlike methods that prune solely based on weight magnitude or perplexity, ALPS incorporates latency ($P_t$) and energy ($P_e$) directly into the comprehensive metric function $s_i$. This allows us to find configurations that fit the same memory footprint as baseline methods but operate much more efficiently.
> >
> > We have revised Lines 44-47 in the final version to more clearly articulate that memory is the prerequisite constraint, while latency and energy are the neglected optimization objectives critical for deployment quality.
> >
> >
> > > Q3: What is the total computational cost (in GPU-hours) of training the encoder–evaluator–decoder model, and how does it compare to conventional pruning search methods in both wall-clock time and energy usage?
> >
> > The Alps framework incurs a one-time, offline computational cost consisting of three stages. Once trained, it generates optimal configurations almost instantly.
> >
> > - Data Collection (Pruning-Score Pairs): The text describes the overhead of constructing the 100 pruning-score pairs (approximately 1 hour, via heuristics and random sampling) as a "modest, one-time cost" that is negligible compared to long-term deployment.
> > - Model Training: The training of the encoder-evaluator-decoder architecture is performed on the collected dataset (approximately 0.5 hour).
> > - Recovery: The post-pruning recovery stage takes approximately 2-3 hours on a single NVIDIA A40 GPU.
> > - Generation Time: After this one-time process, Alps can generate diverse, high-quality pruning configurations for new constraints in under 1 second without further data collection or retraining.
> >
> > The Alps framework replaces the computationally prohibitive search of conventional methods with a modest, one-time offline cost. While exhaustive discrete search suffers from combinatorial explosion (e.g., $101^{32}$ variants for Llama-7B) that makes finding optimal configurations intractable, Alps requires only approximately 1 hour for data collection on a single A40 GPU and 2-3 hours for the post-pruning recovery stage. Once this pre-computation is complete, Alps dramatically outperforms conventional heuristics in adaptability and speed, generating valid, hardware-specific pruning configurations in under 1 second without retraining. Unlike static heuristics that neglect system-level metrics, this minimal upfront investment enables Alps to function as a deployment-specific optimizer, delivering up to 34.1% energy savings and 33.5% latency reduction compared to standard baselines.
> >
> > While Alps incurs a modest upfront cost, it eliminates the prohibitive cost of discrete search and provides superior, energy-efficient configurations instantly for any target device, unlike the static and energy-agnostic results of conventional heuristics.

---

> ### Author Response · Authors · 2025-11-27
>
> Dear Reviewer PaWe,
>
> We are following up to ensure we have fully addressed your specific concern regarding **"Theoretical Clarity"** and the **"Black box"** nature of the latent representation.
>
> In our recent discussion, we have crystallized the theoretical grounding of ALPS, which we believe directly resolves your concern:
>
> **1. Theoretical Grounding:**
> ALPS is an instantiation of the **Latent Space Optimization (LSO)** paradigm. This approach is empirically validated in Neural Architecture Search (**NAO**, NeurIPS 2018 [1]) and shares the core philosophy of **Continuous Relaxation** popularized by **DARTS** (ICLR 2019 [3]).
> * **Response to "Black Box":** The latent space is not an arbitrary black box. It is a **learned manifold** that captures the structural dependencies of the LLM (similar to molecular embedding [2]). The Evaluator acts as a **differentiable surrogate**, enabling gradient ascent to find solutions that discrete heuristics cannot reach.
> * **Commitment:** We commit to incorporating this formal definition and references [1,2,3] into the final paper to ensure the method is theoretically well-positioned.
>
> **2. Addressing "Missing Baselines" & "Hardware Generalization":**
> To demonstrate the effectiveness of this theoretical framework, we conducted extensive new experiments covering **state-of-the-art learned baselines** (AdaPruner, MoDeGPT) and **modern LLMs** (Llama3-8B, Qwen2.5).
> **Results confirm that ALPS achieves better overall trade-offs across these diverse settings**, demonstrating that our "learned manifold" approach generalizes effectively to both advanced pruning baselines and new model families.
>
> We hope this theoretical clarification and the inclusion of the requested baselines (AdaPruner) warrant a reconsideration of your score.
>
> Best regards,
>
> The Authors
>
> **References:**
>
> [1] Luo, R., Tian, F., Qin, T., Chen, E., & Liu, T. Y. (2018). Neural architecture optimization. *Advances in Neural Information Processing Systems (NeurIPS)*.
>
> [2] Gómez-Bombarelli, R., et al. (2018). Automatic chemical design using a data-driven continuous representation of molecules. *ACS Central Science*.
>
> [3] Liu, H., Simonyan, K., & Yang, Y. (2019). DARTS: Differentiable architecture search. *ICLR*.

---

### Official Review · Reviewer_fzwn · 2025-10-29

**Soundness:** 2
**Presentation:** 2
**Contribution:** 2
**Rating:** 4
**Confidence:** 4

**Summary:**

The paper proposes a framework for pruning LLMS, which reformulates the pruning configuration optimization as a continuous optimization problem and solves the problem by gradient-based optimization method.

**Strengths:**

1.	The formulation is novel, which relaxes the LLM pruning problem by using discrete pruning ratio of each layer.
2.	Instead of only focusing on utility, the paper also includes efficiency and latency in the optimization problem.

**Weaknesses:**

1.	The validity of formulation remains questionable. From my point of view, representing a layer by a single pruning ratio is over-simplified. More evidence is needed to prove the simplification works.
2.	The paper claims high efficiency of the proposed method, but the process of collection pruning–score pairs consumes far more energy than other methods.
3.	The continuity and reasonability of the representation space is confused.  The representation space is learned purely from discrete samples of traditional heuristics. Based on that, the gradient optimizer operates on the representation space, with no guarantee that it faithfully reflects the true pruning landscape.

**Questions:**

1. Could you please provide more evidence about the alignment between predicted and the true performance?

2. Could you further explain the choice of using LSTM? From my point of view, the pruning configuration sequence is not temporal; a simple MLP or transformer encoder may also work.

3. As one of the main concern is the efficiency, and the number of heuristic points is highly related to the total cost. Could you please provide how sensitive is ALPS to the number of heuristic configurations used for training?

4. I am quite curious about the stability of Gradient optimization. Does the gradient search in the learned space diverge or produce invalid configurations?

---

> ### Author Response · Authors · 2025-11-26
> **Response to Reviewer fzwn (1/2)**
>
> > W1: The validity of formulation remains questionable.
>
> Sorry for the confusion. We would like to clarify that the single-ratio formulation is a specific design choice that enforces structured sparsity. It is also hardware-compatible. Instead of being a simple on/off switch for layers, the ratio r_i gradually controls the density of structural blocks within each layer. 1) Experimental results demonstrate the effectiveness of ALPS over complex, fine-grained heuristics, which are often used in decision-making processes. 2) As illustrated in Figure 6, ALPS employs non-uniform pruning, preserving sensitive early layers while compressing others. 3) The efficiency improvement offered by this structured granularity results in significant reductions in latency and energy consumption. These improvements far surpass the efficiency gains typically achieved by conventional pruning methods.
>
> > W2: Energy consumption during data collection.
>
> We thank the reviewer for this keen observation. We acknowledge that data collection incurs costs initially, but this is a trade-off for improving the efficiency of edge deployment. Firstly, data collection is a one-time offline process that is only executed during the pre-deployment phase. In contrast, ALPS achieves significant energy savings by reducing inference energy consumption by up to 34.1%, and these savings remain throughout the entire lifecycle. As detailed in Appendix E.2, this offline cost is negligible when amortized against long-term inference. Essentially, ALPS replaces generic 'one-size-fits-all' heuristics with a hardware-aware configuration, ensuring the model is optimised for resource-constrained edge environments.
>
> > W3 & Q1: The alignment between learned representation space and the true pruning landscape.
>
> We appreciate the reviewer’s insightful question. We offer the following strong empirical evidence that the space faithfully captures performance trends:
>
> - Gradient Efficacy: As detailed in Section 4.3 and Figure 5, the full ALPS method (with gradient optimization) significantly outperforms the $Alps^{-G}$ variant (which selects the best discrete sample without optimization). Specifically, gradient optimization yields a 1.91x improvement in generation capability and an 8.15% increase in task performance. If the representation space did not reflect the true landscape, following the gradient would likely degrade performance rather than consistently improve it.
> - Generalization Beyond Training Samples: ALPS consistently discovers configurations that outperform the very heuristic samples used to train it (as shown in Figure 7) . This demonstrates that the evaluator has successfully learned a generalized "scoring surface" that extrapolates correctly to unseen, higher-performing regions, rather than merely memorizing the discrete input points.
>  - Convergence Stability: As discussed in Appendix E.5, the optimization process converged successfully across all experiments without a single failure . This macro-level stability strongly suggests the learned space is smooth and well-behaved, rather than chaotic or misaligned.
> - MSE Minimization: The evaluator is explicitly trained to minimize the Mean Squared Error (MSE) between the predicted score $\hat{s}$ and the ground truth $s$ . The successful convergence of this loss function during training directly quantifies the alignment between the learned predictions and the actual model performance.

---

> ### Author Response · Authors · 2025-11-26
> **Response to Reviewer fzwn (2/2)**
>
> > Q2: Explain the choice of using LSTM?
>
> We selected the LSTM-based Seq2Seq architecture to align with the structural nature of LLMs and optimize efficiency:
> Sequential Layer Dependencies While not temporal, LLM layers are computationally sequential (Input $\to$ Layer 1 $\to$ ...). Pruning an earlier layer alters feature propagation to all subsequent layers. Unlike MLPs, which treat configurations as independent flat vectors, LSTMs naturally model these cascading dependencies, conditioning the pruning ratio for layer $i$ on the decisions for layers $1 \dots i-1$.
>
> Efficiency vs. Transformers Given the short sequence length (e.g., $k=32$ for Llama-7B), Transformers introduce unnecessary complexity. As detailed in Appendix E.6, our experiments showed that Transformers increased training costs and data requirements without yielding performance gains over the simpler, more efficient LSTM.
>
> > Q3: How sensitive is ALPS to the number of heuristic configurations used for training?
>
> We appreciate this important question regarding the cost efficiency of our data collection. We have conducted a specific sensitivity analysis (Section 4.3, Figure 7) to demonstrate that ALPS is robust and cost-effective.
>
> - **Data efficiency via augmentation and diversity.** First, ALPS is designed to be highly data-efficient. As detailed in Section 3.1, we applied a 10x Mixup data augmentation to expand the collected configurations for training. This allows us to learn a smooth representation space without requiring a massive number of expensive raw evaluations.
> Furthermore, as discussed in the "Study of Generalization Ability" (Section 4.3 and Figure 7), the strength of ALPS lies in the diversity of the knowledge base rather than just quantity. The goal is to learn a generalized representation of "what makes a good pruning configuration" from different "experts". Our results show that combining diverse pruning methods (ALPS) significantly outperforms strategies relying on single heuristics, emphasizing that a diverse, small set is more valuable than a large, homogeneous one.
>
> - **Sensitivity analysis of heuristic points.** To directly address your concern, we conducted a sensitivity analysis on the number of heuristic configurations. Fixing the random samples at 50, we varied the number of heuristic samples across $\{0, 10, 30, 50, 80\}$.
> We observed that increasing the number of heuristic samples (e.g., from 0 to 30) leads to better learning foundations and improved final decision-making, confirming that expert knowledge guides the search space effectively. However, the performance gains tend to saturate after reaching approximately 30-50 heuristic points. We attribute this convergence to the fact that the key patterns of layer redundancy (e.g., which layers are universally critical) are learned relatively quickly. Once the core structural dependencies are captured by the initial diverse set of heuristics, adding more samples provides diminishing marginal information gains. The 10x Mixup augmentation further fills the continuous space between these points, rendering additional raw samples unnecessary. Thus, our choice of 50 heuristic points represents a cost-effective Pareto optimum, ensuring high performance without incurring excessive collection costs.
>
> > Q4:  The stability of Gradient optimization.
>
> We appreciate the reviewer’s insightful question. ALSP address the concern regarding optimization stability by:
> First, the gradient search in the learned space has proven empirically stable; we recorded zero divergence across all runs (Appendix E.5), confirming that the representation space supports smooth optimization using Adam. Second, validity is structurally guaranteed. Because the decoder treats pruning ratios as a fixed vocabulary in $[0, 1]$ and strictly enforces a sequence length matching the layer count $k$, the model is physically incapable of generating invalid configurations. Finally, to ensure the search yields high-quality outcomes and escapes local minima, we utilize a multi-start strategy initialized from the Top-K ($K=25$) candidates found during the offline collection phase.

---

> ### Author Response · Authors · 2025-11-27
>
> Dear Reviewer fzwn,
>
> We wanted to share a final update regarding your concern about the **"validity of formulation"** (specifically, whether the single pruning ratio is over-simplified).
>
> **1. Why the Formulation Works (The Manifold Perspective):**
> While representing a layer by a ratio seems simple, our method operates on the **Latent Space Optimization (LSO)** principle. Similar to **NAO [1]** and **DARTS [2]**, the core complexity is handled by mapping discrete decisions into a continuous **manifold**.
> * **Key Insight:** This allows ALPS to capture complex, non-linear dependencies between layers in the latent space, even if the output format is a simple ratio sequence. The complexity is handled by the **learned representation**, not the manual heuristic.
>
> **2. Strong Empirical Proof:**
> To **demonstrate** that this formulation is not "too simple," we significantly expanded our evaluation to include **modern models** (Llama3-8B, Qwen1.5, Qwen2.5) and **complex learned baselines** (AdaPruner, MoDeGPT).
> **ALPS achieves better overall trade-offs in these rigorous comparisons**, confirming that our formulation successfully captures critical redundancy patterns even when challenged by the latest architectures and SOTA methods.
>
> We hope these new extensive results demonstrate the robustness of our approach.
>
> Best regards,
> The Authors
>
> **References:**
>
> [1] Luo, R., Tian, F., Qin, T., Chen, E., & Liu, T. Y. (2018). Neural architecture optimization. *Advances in Neural Information Processing Systems (NeurIPS)*.
>
> [2] Liu, H., Simonyan, K., & Yang, Y. (2019). DARTS: Differentiable architecture search. *International Conference on Learning Representations (ICLR)*.

---

### Meta-Review · Area_Chair_ZCze · 2026-01-01

**Summary:**

The paper proposes ALPS, an adaptive LLM pruning framework that reformulates configuration search as a gradient-based optimization problem within a learned latent space. However, after reviewing the paper and author-reviewer discussions, I find the current version still holds unresolved concerns regarding its theoretical clarity and formulation, making it not yet ready for acceptance.

---

The authors have recognized that the reviews from Reviewer PaWe contain factual hallucination, which was confirmed by the AC. Therefore, the AC downgraded Reviewer PaWe's review as this is a clear misconduct according to the reviewer guideline.

---

In addition, this paper was flagged due to possibly hallucinated references for:
* Statista Inc. Mobile ram usage worldwide from 1q-19 to 1q-21 (in gb per device). , 2021.

After carefully double-checking, **the AC took this as outdated data rather than a hallucination**: Statista is a company that dynamically releases mobile statistics databases each year or quarter. Though the data between 1q-19 to 1q-21 cannot be retrieved currently, it is highly likely that the original page with that exact title is archived or updated.

The authors are reminded to update the reference and provide the URL link in the updated paper.

**Reviewer Concerns:**

**Addressed Concerns**:
1. **Missing Baselines and Models (Reviewer onua)**: The authors successfully addressed concerns regarding the lack of modern comparisons by adding experiments with Llama3-8B, Qwen1.5, and Qwen2.5, and comparing against stronger baselines like AdaPruner, MoDeGPT and SVD-LLM in their rebuttal.
2. **Efficiency and Cost Analysis (Reviewer fzwn)**: The authors provided a detailed breakdown of the offline computational overhead (e.g., 1 hour for data collection, ~0.5 hours for training, and 2–3 hours for recovery on a single NVIDIA A40 GPU). Given the reported 33.5% reduction in inference latency, this one-time offline cost is considered justifiable.

**Outstanding Concerns**:
1. **Theoretical Clarity and Formulation (Reviewer fzwn, Reviewer onua in his/her second review)**: Reviewer fzwn pointed out that characterizing a layer's pruning configuration solely by a single pruning ratio is oversimplified, and noted a lack of evidence demonstrating the continuity and reasonability of the representation space. Although the authors frame ALPS as an instantiation of Latent Space Optimization akin to NAO or DARTS in their rebuttal, they did not provide sufficient theoretical justification or rigorous analysis to explain how a structurally valid and smooth latent manifold can be effectively learned from such sparse heuristic samples. This theoretical deficiency remains a critical obstacle to the paper's acceptance.

**Reviewer Scores:**

* Reviewer fzwn: Score likely remains unchanged (4). While empirical baselines were enriched, the reviewer's fundamental doubts regarding the theoretical soundness of the formulation remain unresolved.
* Reviewer PaWe: the AC is leaning to downgrade/disregard this review due to substantiated evidence of factual hallucinations, which is a clear misconduct due to the reviewer guideline.
* Reviewer onua: Score likely remains unchanged (6). The reviewer explicitly stated that the rebuttal addressed most concerns, but remained hesitant to increase the score further considering the theoretical clarity.

---

### Decision · Program_Chairs · 2026-01-26

Reject